# Systematic Security Analysis of Sensors and Controls in PV Inverters: Threat Validation and Countermeasures [note 1]

**DOI:** 10.3390/s25051493

**Published:** 2025-02-28

**Authors:** Fengchen Yang, Kaikai Pan, Chen Yan, Xiaoyu Ji, Wenyuan Xu

**Affiliations:** Zhejiang University, Hangzhou 310027, China; yangfengchen@zju.edu.cn (F.Y.); yanchen@zju.edu.cn (C.Y.); xji@zju.edu.cn (X.J.); wyxu@zju.edu.cn (W.X.)

**Keywords:** power inverter, sensors, electromagnetic interference, countermeasures

## Abstract

As renewable energy sources (RES) continue to expand and the use of power inverters has surged, inverters have become crucial for converting direct current (DC) from RES into alternating current (AC) for the grid, and their security is vital for maintaining stable grid operations. This paper investigates the security vulnerabilities of photovoltaic (PV) inverters, specifically focusing on their internal sensors, which are critical for reliable power conversion. It is found that both current and voltage sensors are susceptible to intentional electromagnetic interference (IEMI) at frequencies of 1 GHz or higher, even with electromagnetic compatibility (EMC) protections in place. These vulnerabilities can lead to incorrect sensor readings, disrupting control algorithms. We propose an IEMI attack that results in three potential outcomes: Denial of Service (DoS), physical damage to the inverter, and power output reduction. These effects were demonstrated on six commercial single-phase and three-phase PV inverters, as well as in a real-world microgrid, by emitting IEMI signals from 100 to 150 cm away with up to 20 W of power. This study highlights the growing security risks of power electronics in RES, which represent an emerging target for cyber-physical attacks in future RES-dominated grids. Finally, to cope with such threats, three detection methods that are adaptable to diverse threat scenarios are proposed and their advantages and disadvantages are discussed.

## 1. Introduction

Renewable energy sources (RES), e.g., solar, wind, or hydroelectric power, are replacing fossil fuels to reduce their impact on global climate change [1] and have been reported to account for 30% of all energy sources up to 2023 [2]. As the penetration rate of RES continues to increase, it is critical to examine the emerging security issues of the power grids before RES constructions are finalized. Since most RES generates direct current (DC) power, yet the grids and power consumers operate on alternating current (AC) power, millions of power inverters have to be installed to convert DC power into AC power for each RES, as shown in Figure 1. Thus, the security of power inverters can affect the smooth operation of RES power generation and even the stability of the power grids.

Building on our previous conference paper [3], we present a more detailed analysis of the intentional electromagnetic interference (IEMI) threats to photovoltaic (PV) inverters (also called solar inverters) and propose potential countermeasures. The goal is to provide valuable security insights for device developers and designers. As one of the most important renewable energy sources, new solar capacity added between now and 2030 will account for 80% of the growth in renewable power globally by the end of this decade [2].

In this paper, we focus on the distinct security of inverters, i.e., the threat of intentional electromagnetic interference (IEMI) on the analog sensors of power inverters, since inverters rely on the correct sensing of voltage and current of input power sources as well as the grids to ensure stable and safe power conversion. For instance, without accurate sensing of current and voltage, the inverter may fail to detect islanding conditions (when the grid is down but the inverter is still producing power) and potentially cause fires or electrocute a maintenance technician [4]. Although modern electronic devices are typically designed to withstand normal electromagnetic interference (EMI), recent studies have demonstrated that Intentional Electromagnetic Interference (IEMI) poses significant new threats to sensors [5,6,7,8]. IEMI involves the malicious generation of electromagnetic signals aimed at disrupting or damaging electronic systems. Jie et al. [9] have extensively analyzed the conducted and radiated susceptibility of power electronics to IEMI, highlighting its potential risks. Most PV inverters are installed in unguarded areas, e.g., resident backyards, building rooftops, or power plants in a desert [10], whereby immersing sensors with malicious IEMI signals is possible.

These observations motivate us to perform further investigation into the impact of IEMI on PV inverters, yet the DC–AC power conversion circuits inside inverters generally handle 50 watts up to 50 kilowatts [11] and are a natural and strong source of IEMI by design. For instance, power semiconductor switches that commutate at high switching frequencies will radiate IEMI. Thus, all power inverters have to satisfy the electromagnetic compatibility (EMC) requirements by properly grounding, adding filters, and shielding so that they can operate normally in the presence of self and mutual interference. Although prior work [12] has shown that a static magnetic field can affect Hall sensors at a distance of 10 cm, it is unclear whether an IEMI injection could affect other types of embedded sensors, e.g., voltage sensors, and whether IEMI signals can be crafted to precisely manipulate chosen sensors, as well as their consequences on inverters as a whole.

In this work, we performed a systematic security analysis of the PV inverters on real inverters and microgrid (microgrid is a mini version of the grid, where it contains a group of interconnected loads and distributed energy resources and can connect to the grids or operate in an islanding mode [13,14]), we find that both the embedded current and voltage sensors in PV inverters are vulnerable to IEMI, although they conform to EMC standards on conduction and radiation interference [15,16,17].

In general, EMC includes both EMI and electromagnetic susceptibility (EMS). Intentional electromagnetic interference (IEMI) is considered part of EMS because it intentionally targets and exploits the vulnerabilities of a system. Unlike unintentional EMI, which results from external noise, IEMI deliberately interferes with the system’s electromagnetic environment, causing disruptions in its operation.

We believe that three reasons cause such vulnerabilities. First, the EMC is designed to cope with unintentional electromagnetic interference (EMI), and its frequency band does not cover the range of intentional electromagnetic interference (IEMI). The EMC standard mainly considers two types of EMI: the conducted interference in the range of 0.15MHz∼30MHz and the radiated interference in the range of 30MHz∼1GHz [18,19,20]. Yet, IEMI signals around or higher than 1GHz may be able to bypass the EMC measures. Second, although low-pass filters are meant to remove all interference signals with a frequency higher than 0.15MHz, the real filters are not ideal and can let go of high-frequency signals, such as a phenomenon that has long been recognized in academia, filter leakage [21,22,23,24]. Lastly, it is worth noting that certain inverter designs may inadvertently introduce vulnerabilities to IEMI. For instance, ① the presence of an LCD screen in the inverter may create a gap in EMC protection, providing a potential entry point for IEMI; ② non-ideal alignment of the printed circuit board (PCB) and device layout can result in parasitic capacitance; ③ the asymmetrical arrangement of circuits on the PCB can compromise the inverter’s immunity to common-mode interference; ④ the control algorithms of PV inverters often rely on the assumption that sensor measurements are both reliable and consistent, without sufficient checks in place, which can allow false voltage and current measurements to deceive the control system. While parasitic capacitance remains a common issue in most medium-voltage power electronic converters [25], current research primarily focuses on predicting and mitigating this effect [26,27,28,29,30]. However, many of the proposed methods tend to increase material and manufacturing costs [25].

To illustrate the impact of the aforementioned vulnerabilities in combination, we propose three types of consequences on PV inverters by emitting carefully crafted IEMI, as shown in Figure 1.

DoS: The PV inverter shuts down completely, causing an instantaneous power reduction in PV generation to the grid or consumers.Damage: The PV inverter can be physically burned out and has to be repaired or replaced.Damping: This type of threat causes the output power of PV inverters to be lower than their capability. Long-term continuous Damping will reduce the efficiency of the PV generation.

We have validated the consequences of an IEMI attack on a PV inverter development kit, six single-phase and three-phase commercial kilowatt-level PV inverters, and a rural-scale microgrid operated in the real world, by transmitting IEMI signals at a distance of 100∼150cm and emission power within 20W. Despite the fact that the power capabilities of PV inverters vary from a few kilowatts to 60 kilowatts, the embedded current and voltage sensors operate on a voltage level of 5V and are all vulnerable to IEMI signals. We have uploaded video demonstrations to the link (https://tinyurl.com/ReThinkDemoVideos, accessed on 20 February 2025).

To enhance the security of PV inverters, we investigate the root causes of IEMI threats and propose three detection methods from three levels. (1) From the signal level, we propose a detection method leveraging the distributed effect of IEMI. (2) From the model level, we introduce a detection method based on the energy conservation law. (3) From the combination level, we present a detection method utilizing neural networks. Then, we evaluate the effectiveness of these methods on the Ti C2000 PV inverter, analyze potential influencing factors, and provide a comparison of their characteristics. We hope our work provides valuable insights for designing active defenses against IEMI threats in PV inverters.

To the best of our knowledge, this is the first systematic work analyzing the impact of IEMI on PV inverters and validating the real-world microgrid. Our work is complementary to existing studies on traditional software or communication-related issues, e.g., software vulnerabilities of inverters or DoS and replay attacks against DC microgrids [31,32,33,34,35]. The goal of our work is to raise awareness of the security of power electronic devices in the power grids as RES are increasingly being adopted and they represent an emerging Cyber-Physical Systems (CPS) threat surface. We imagine that our analysis and conclusions may potentially lay the groundwork for analyzing other types of inverters and power electronic devices with similar sensors and control logic. In summary, our contributions are as follows:We present a systematic security analysis of PV inverters and analyze the vulnerabilities of sensors and control algorithms susceptible to IEMI signals.We illustrate the adversarial scenarios that can shut down, permanently damage, and dampen the power output of PV inverters, and we validate the threat on commercial PV inverters and a real-world microgrid.We investigate the underlying causes of these vulnerabilities and propose three effective detection methods to counter these threats.

## 2. Related Works

This section provides an overview of the existing works, focusing on three aspects: the security of power converters and the defense strategies against IEMI attacks. By analyzing these works, we aim to identify research gaps and highlight the contributions of this study in addressing the new threats.

### 2.1. Security of the Power Converters

Existing security research on power converters mainly focuses on the digital world, e.g., Liu et al. (2015) studied false data injection (FDI) attacks on power grid state estimation and proposed detection methods [36]. In recent years, analog-world attacks have been proven to be a new type of FDI attack against the power grid. Barua et al. [12] investigated a magnetic field-based attack called Hallspoofing on inverters. This attack manipulates the Hall current sensor by placing an electromagnet next to the inverter, potentially causing the inverter to burn out or shut down. The distinctions between Hallspoofing and our work are described as follows: ① Hallspoofing is limited to manipulating Hall current sensors, whereas our work addresses the threat of IEMI on both Hall and non-Hall sensors. Notably, non-Hall sensors in inverters may render Hallspoofing impractical for precise manipulations. ② Due to the constraints of the magnetic field, the attack distance of Hallspoofing is restricted to a few centimeters. ③ In contrast to Hallspoofing, our analysis is comprehensive, delving into vulnerabilities within the inverter’s control algorithms. ④ We revealed a previously unrecognized threat that can directly result in irreversible physical damage to the inverter. Note that achieving Damage involves targeting the DC bus voltage sensor, which is distinct from Hall current sensors.

### 2.2. Countermeasures Against IEMI Attacks

Current strategies for defending against IEMI threats encompass both passive and active approaches. Passive defenses primarily involve the use of shielding [37,38,39,40] and filtering techniques [41,42,43]. On the other hand, active defenses focus on detection mechanisms, which can be categorized into the following methods: ① incorporating additional detection circuits to monitor EMI [44,45,46,47,48], ② applying secret encoding to critical signals to identify IEMI presence [49,50,51], and ③ developing detection algorithms based on the intrinsic characteristics of sensors [52,53,54,55,56].

In summary, passive defenses can directly mitigate IEMI threats by introducing additional hardware, but they are more costly and have limitations due to components’ physical vulnerabilities (e.g., filter leakage). Of course, many recent works have made efforts in characterization, impedance modeling, and impedance measurement to enhance the filtering effect [57]. Conversely, active defenses are more cost-effective in mitigating the impact of IEMI attacks while providing timely alerts. We believe that the hybrid of passive and active defenses will generate a better effect. Our work proposes three detection methods based on the inverter efficiency feature, the amplitude features of IEMI noise, and hybrid features extracted by the neural network, which indicates a promising direction for future research.

## 3. Background and Threat Model

### 3.1. Principle of PV Inverter

PV inverters, like many other types of inverters, are the heart of every PV system. To satisfy various design requirements, PV inverters may have subtle differences in their circuit design [58]. After examining 47 inverters from three leading manufacturers [59,60,61], we found that 43 inverters employ a standard DC–DC–AC topology and this predominant architecture is known as a Two-Stage Power Conversion (TSPC) system [62], which is the focus of this paper. Particularly, a PV inverter consists of a power conversion unit, multiple current and voltage sensors, and control algorithms. Since power generation efficiency is one of the most important goals, a PV inverter will track the PV panel’s maximum power point (MPP) by sensing and incorporating various control algorithms to convert DC power into AC power. To understand the details, we introduce them below.

#### 3.1.1. Power Conversion Unit

A typical TSPC PV inverter contains two parts: the DC–DC stage and the DC–AC stage, as shown in Figure 2.

DC–DC Stage. The primary function of the DC–DC stage is to increase the voltage level from the PV panel output, e.g., ranging from 30V to 60V, to the one required by power grids, i.e., 325V peak for the single-phase and 565V peak for three-phase.

DC–AC Stage. The DC–AC stage converts the direct current on the DC bus to the AC that can be fed into the grid through the inverter circuit, with the help of two control algorithms, i.e., voltage control loop and current control loop.

#### 3.1.2. Control Algorithm

The PV inverter relies on control algorithms to maintain the PV panels or arrays working at their maximum power state and convert DC into AC for integration into the grid. There are three main parts: the maximum power point tracking (MPPT) algorithm, the voltage control loop, and the current control loop.

MPPT Algorithm. To maintain the highest energy conversion efficiency in various atmospheres [63,64], the MPPT operates along a voltage-current (V-I) curve to identify the maximum power point (MPP), where the V-I curve is an inherent characteristic of the PV panel and varies with the irradiance and temperature. The most commonly used MPPT algorithm is the Perturb and Observe (P & O) method, where the basic idea is to try adding a perturbation to the inputs of PV inverters and measure the resulting power [65].

Voltage Control Loop. The role of the voltage control loop is to adjust the DC bus voltage $V_{dc}$ to a reference value. The DC bus capacitor functions as an energy buffer to stabilize the DC bus voltage. If the input power exceeds the output power, the capacitor $C_{dc}$ on the DC bus will continue to be charged, which will lead to an increase in $V_{dc}$ and trigger the voltage control loop to raise the output reference current $I_{dref}$. Before entering the PI control, the coordinate system transformations (Clarke and Park) [66] are applied to the measured three-phase voltage and current.

Protection Mechanism of PV Inverter. In the operation of PV inverters, a set of self-protection mechanisms are incorporated to prevent safety issues that may arise from device damage and circuit failure. The mechanisms considered in this paper include DC bus over-voltage protection, as well as AC over and under-voltage protection [67].

DC bus over-voltage protection. The PV inverter continuously monitors the voltage of the DC bus. If the DC voltage exceeds a predefined threshold several times, the inverter disconnects from the grid and stops power generation.AC over and under voltage protection. When the inverter’s output voltage is detected to be higher than the threshold range, it will disconnect itself from the grid. If the output voltage drops outside the allowable range of low voltage crossing (20%), the low voltage crossing function will activate, triggering an alarm.

### 3.2. Sensors of PV Inverter

As illustrated in Figure 2, PV inverters rely on embedded sensors to measure voltage and current and feed them back to the control loop.

#### 3.2.1. Non-Hall Voltage Sensor

Voltage sensors convert hundreds of volts into a few volts that the analog-to-digital conversion (ADC) module can handle. Besides, since inverters operate in complex electromagnetic environments and tend to generate common mode noise in the circuits, differential operational amplifiers (op−amp) are often employed to suppress noises [68]. A typical structure of a differential op−amp circuit is shown in Figure 3a, and the magnification can be expressed as follows Equation (1):(1)$$u_{o}=\frac{R_{3}\cdot \left(R_{1}+R_{F}\right)}{R_{1}\cdot \left(R_{2}+R_{3}\right)}\cdot u_{i2}-\frac{R_{F}}{R_{1}}\cdot u_{i1},$$
where the $u_{i1}$ and $u_{i2}$ are the inverted and in-phase input signals, $u_{o}$ is the output signal, $R_{1}$ and $R_{2}$ are the input resistors, $R_{F}$ is the feedback resistor and $R_{3}$ is the ground resistor. The magnification is determined by the resistors of the op−amp. In practice, resistors $R_{1}$ and $R_{2}$ usually consist of multiple divider resistors in series, and they step down the high voltage to a low voltage signal within 5V; thus, for inverters from a few kilowatts to hundreds of kilowatts, the embedded voltage sensors shall be vulnerable to IEMI signals at similar power levels.

#### 3.2.2. Hall Current Sensor

Inverters typically use a Hall current sensor, which converts the magnetic field generated by the current into DC or AC voltage based on the Hall effect [69]. As shown in Figure 3b, the current *I* generates a magnetic field *B*, and *B* is proportional to *I* according to Ampere’s Law. Then the electrons moving on the electrode plate will be subjected to the Lorentz force $F_{L}$ in B and move to the sides of the electrode plate, and generate an electric field *E* on the electrode plate. Finally, a balance state will be reached when the electric field force and the Lorentz force are equal, which can be formulated as Equation (2), where *d* is the width of the electrode plate and *q* is the electrical charge. Since *B* is proportional to *I* and $V_{H}$ is proportional to *B*, the Hall sensor’s output $V_{H}$ is proportional to the current *I*. Finally, Hall current sensors use a similar op−amp to suppress the common-mode noise in $V_{H}$ and output the measurement result.(2)$$B\cdot q\cdot v=q\cdot E=q\cdot \frac{V_{H}}{d}$$

### 3.3. Threat Model

This manuscript is an expanded version of ReThink [3] and applies the same threat model as it does. We make the following assumptions about the adversary:

Attack Goal. The attacker’s goal is to covertly cause the shutdown, power reduction, or even burnout of a PV inverter. Though ambitious attackers may target a group of inverters and try to create potentially escalated impacts such as voltage or frequency fluctuations or even blackouts in a local microgrid, we focus on basic attacks against individual inverters in this paper.

Non-contact Access. We assume the attacker can approach the target inverters within a few meters, but they cannot physically touch or damage them due to safety and stealthiness concerns. Alternatively, the adversary can leave a camouflaged IEMI device nearby and control it remotely.

Prior Knowledge. We assume that adversaries could have prior knowledge of the target inverter. Given that many PV inverters are commercial products readily available on the market, the adversary could acquire a PV inverter of the same model and conduct necessary tests beforehand. More favorably, in practice, PV systems in a region often use the same model of PV inverters.

## 4. Understanding the Impact of IEMI on Embedded Sensors of PV Inverters

In this section, we explore how IEMI affects embedded voltage sensors and current sensors of PV inverters through theoretical analysis and feasibility experiments.

### 4.1. Analysis of the IEMI Impact on Sensors

#### 4.1.1. Impact of IEMI on Voltage Sensors

The sensor’s PCB usually carries parasitic capacitance and is susceptible to electromagnetic interference in the environment. Besides, the op−amp circuit will further rectify and amplify the coupled signals. The transmission process can be illustrated in Figure 4a, and there are four steps:

EMI signal injection. Process ① in Figure 4a is IEMI injection. Electromagnetic fields around the sensor can be injected into sensor circuits (e.g., input nodes) via electromagnetic coupling. Generally, according to the IEMI transmission paths, IEMI coupling methods can be divided into conductive coupling, inductive coupling, capacitive coupling, and radiative coupling (also called radio frequency interference, RFI) [70,71]. Among them, radiative coupling refers to the far-field coupling of higher-frequency signals in the microwave frequency range, which can be transmitted over longer distances. Notably, the conductors (e.g., copper wires and component pins) and the insulator (e.g., PCB substrate) on the sensor’s PCB will form parasitic capacitance, as shown in Figure 4b. These parasitic capacitances are susceptible to the aforementioned high-frequency electric fields, which can introduce interfering signals.Nonlinear rectification effect. The amplifier can rectify the high-frequency AC signal at the input and generate a DC bias at the output. The main reason is that the bipolar junction transistor (BJT) in the op−amp chip contains p-n junction diodes, which are efficient rectifiers due to their nonlinear current–voltage characteristics, especially in low-power op−amps [72]. When a high-frequency signal v(t)=VXcos(2πfXt) is injected into the base-emitter junction of an op−amp BJT-based input stage, the output will generate an AC term ΔiC(AC) at twice the input frequency and a DC term ΔiC(DC) [72], which can be described by Equation (3):(3)ΔiC(DC)=(VXVT)2·IC4,
where VX is the amplitude of the noise signal and VT is the thermal voltage of the transistor, which is relative to the temperature.Asymmetric differential effect. The asymmetric design of the op−amp circuit on the PCB allows the output bias of the op−amp to be positive or negative. As shown in Figure 5, an op−amp channel consists of a differential amplification input stage, an intermediate amplification stage, and a push–pull output stage. The transfer relationship of the differential amplification input stage can be expressed as follows:Vo1−Vo2=Ad(Vi1−Vi2)+Ac(Vi1+Vi2)≈Ad(Vi1−Vi2)
where Ad is the differential-mode gain and Ac is the common-mode gain.Figure 5The structure of the OPA2171 used in voltage and current sensors.
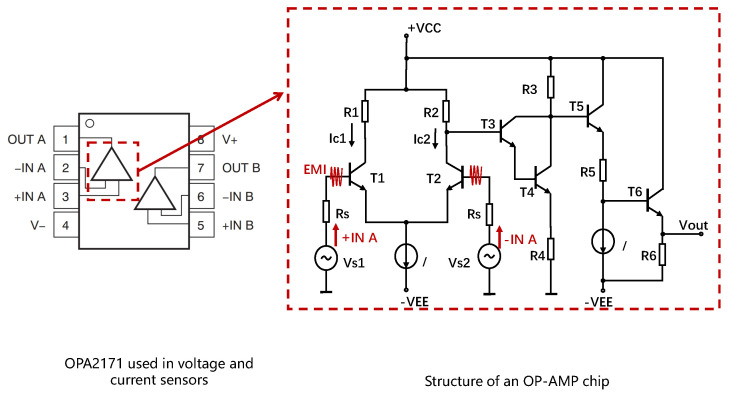


The asymmetric design of the input stage’s wires results in different frequencies of IEMI coupling. Consequently, the IEMI signals coupled into $V_{i1}$ and $V_{i2}$ will differ, ultimately producing a positive or negative output. This outcome depends on whether the coupled signal is stronger at $V_{i1}$ and $V_{i2}$. To demonstrate, we build the circuit model of the OPA2171 chip in Simulink and we inject the sinusoidal signal in Figure 6a to $V_{i1}$, $V_{i2}$ or both, and we find that the output can be positive, negative, or 0, respectively, as shown in Figure 6b–d. Therefore, the attacker can tamper with the sensor’s output to a larger or smaller value by adjusting the frequency of the IEMI signal.

Amplification effect. Amplification is the fundamental function of op−amp. Signal inputs will be amplified according to the set gain; however, IEMI signals can enter into various nodes via radiative coupling. As shown in Figure 3a, when the IEMI signal is injected into the node *b*, it can be considered that $R_{1}=R_{2}=0$. Then, according to Equation (1), the gain will be abnormally large. In other words, even if injecting a millivolt signal at node *b*, it can be amplified to a few volts in process ③ of Figure 4a.

In conclusion, electromagnetic coupling enables the injection of EMI, the nonlinear rectification converts alternative interference into positive bias, the asymmetric differential effect allows the bias to be positive or negative, and the amplification effect amplifies the injected IEMI signals.

#### 4.1.2. Impact of IEMI on Current Sensors

Unlike the voltage sensor, the current sensor includes not only an op−amp circuit but also a Hall element, which may serve as a new entrance for EMI. Thus, we mainly analyze how IEMI can enter the sensor circuit through the Hall chip.

We have already described that Hall current sensors measure current indirectly by measuring the magnetic field generated by the current, and the measurement relies on the balance of the Lorentz force and electric field force on the electrons, as shown in Equation (2). Thus, an additional magnetic or electric field around the Hall chip will impact the current measurement, either directly or indirectly. Now we discuss them separately:Impact of magnetic field on Hall sensor. We assume the measured current generates a magnetic field *B* in the Hall element. Since the output $V_{H}$ is proportional to *B*, we quantify this as Equation (4). If IEMI generates a magnetic field $B_{A}$ nearby, $B_{A}$ will be superimposed on *B*. Therefore, the output of the Hall element may be directly manipulated by the IEMI signal, and this relation can described as Equation (5), and the output $V_{H}$ of the Hall element will be changed by $k\cdot B_{A}$.(4)$$V_{H}=k\cdot B$$(5)$$V_{H}^{*}=k\cdot \left(B+B_{A}\right)=V_{H}+k\cdot B_{A}$$Impact of electric field on Hall sensor. According to Equation (2), we have(6)$$V_{H}=d\cdot E$$

If an additional electric field $E_{A}$ exists near the Hall chip, at this point we have(7)$$V_{H}^{*}=d\cdot \left(E+E_{A}\right)=V_{H}+d\cdot E_{A}$$

Thus, the output $V_{H}$ of the Hall chip will be changed by $d\cdot E_{A}$, where *d* is the width of the electrode plate.

Then, the affected output $V_{H}^{*}$ will continue to be rectified and amplified by the op−amp and finally generate a bias on the measurement, as shown in ③ and ④ in Figure 7.

It is worth noting that since the output $V_{H}$ of the Hall chip is fed into the positive input of the op−amp, the IEMI injected into the Hall chip will theoretically result in a positive bias in the current measurement. However, the IEMI can also affect the op−amp of the current sensor, which will cause positive or negative bias.

### 4.2. Experimental Verification

To verify the previous analysis, we conducted feasibility tests to explore the capability of IEMI to impact sensors of PV inverters.

#### 4.2.1. Can IEMI Impact Voltage and Current Sensors

We conduct an IEMI frequency sweep test on voltage and current sensors. The experiment setup is shown in Figure 8, and the test steps are as follows:

① In the feasibility verification stage, we built the PCBs of the voltage and current sensors according to the schematic of the C2000 PV inverter from Texas Instruments (TI) that we have in hand [73], as shown in Figure 9a,b. ② We use a DC power source RIGOL DP711 [74] to generate a 30V voltage and 0∼5A current to be measured. Then, we use the Arduino UNO to read the voltage every 10ms and send the data to the PC through the serial port. The Arduino is wrapped in EM shielding material to prevent EMI. All components are readily available on the market. ③ Subsequently, we use EXG vector signal generator [75] to generate a 700MHz∼2.5GHz signal, use amplifier HPA-50W-63+ [76] to amplify it to 10W, and emit it with a 5G directional antenna [77] with +14dBi at a distance of 50cm.

We record the deviation of the measurements in Figure 10. For the voltage sensor, the measured voltage can be decreased by 200V and increased by 120V at most. For the current sensor, the measured current can be increased by up to 320A and decreased by up to 30A. The result demonstrates that IEMI can effectively affect the voltage and current sensor’s outputs. Notably, in the test of the Hall current sensor, the deviation in the measurement is predominantly positive. This verifies our previous analysis of the impact of IEMI on current sensors.

To further verify that the IEMI can impact the Hall chip directly, we conducted a small test: We measured the output $V_{H}$ of the Hall chip using RF wines to avoid wire coupling, with the sample rate of 10GHz, and compare the effect of IEMI on $V_{H}$. The result shows that IEMI can directly impact the Hall chip by inducing a 0.2V bias and a 0.5V oscillation on the output $V_{H}$ of the Hall element.

#### 4.2.2. Whether the Impact Is Controllable

To explore the IEMI manipulation capability on sensors, we tested two kinds of IEMI signal modulation methods:

① Frequency modulation (FM). Figure 10a,b reveals that sensors have different “sensitivity” to IEMI signals of various frequencies. It appears that adjusting the signal frequency may manipulate the target sensor’s output. However, we can also find that the sensor’s output varies significantly as the frequency changes. Therefore, achieving precise control of sensor values with FM proves challenging.

② Amplitude modulation (AM). Another signal modulation method is AM, as described in Equation (8), where $s_{m}\left(t\right)$ is the modulation signal, and $A_{c}$ and $f_{c}$ are the amplitude and frequency of the carrier signal $s_{c}\left(t\right)$.(8)$$s_{AM}\left(t\right)=A_{c}\left[1+s_{m}\left(t\right)\right]cos2\pi f_{c}t$$

Since the offset of the sensor’s output is proportional to the amplitude of the IEMI signal, we first select a carrier signal $s_{c}\left(t\right)$ that can impact the sensor’s output, and then set $s_{m}\left(t\right)$ to the “desired” curve, which is also the envelope of $s_{AM}\left(t\right)$.

In this scenario, assuming that one wants the measured voltage to first increase or decrease and then change as the triangular or sine wave, we conducted an experiment using AM. The result is highly “favorable” for an adversary, as depicted in Figure 11. Although the real voltage or current remains constant, the measured values change precisely by the $s_{m}\left(t\right)$, such as triangular and sine waves.

#### 4.2.3. Verification of the Universality and Extensibility

Commercial PV inverters usually contain multiple types of sensors. To analyze the universality of the threat, we propose two questions: ① What is the impact of IEMI on different Hall sensors? ② If there are multiple sensors, can IEMI only impact a single target sensor or control multiple target sensors simultaneously?

Universality. To answer the first question, we evaluate the impact of IEMI on seven different Hall sensors, including four analog sensors and three digital sensors. Hall digital sensors include a speed sensor, a north pole sensor, and a water flow sensor. The result is presented in Table 1. We can find that both wired and wireless Hall current sensors are susceptible to EMI, and wireless Hall current sensors exhibit a higher degree of susceptibility. Hall sensors with digital outputs, like speed sensors, may experience bit-flipping under EMI.

Extensibility. Since IEMI signals of different frequencies can be injected into different nodes of the victim circuit, we can establish a frequency sweep model for each sensor and implement the following: ① “one-to-one” manipulation: select a frequency that exclusively affects the target sensor without impacting others; ② “many-to-many” manipulation: when manipulating several sensors simultaneously, owing to the superposition of IEMI signals, we can employ different channels to emit IEMI signals of various frequencies. This feature also highlights one of the advantages of IEMI over constant magnetic field attacks in Hallspoofing [12]: higher extensibility in signal design through signal multiplexing.

## 5. Understanding the Impact of Sensor Spoofing on PV Inverters

Here, we analyze how the spoofing of sensors affects the operation of PV inverters. We build the PV inverter circuit model and implement the control algorithms outlined in Section 3 using Simulink.

### 5.1. Impact of DC Bus Voltage Sensor

Deceiving the DC bus sensor will directly affect the DC bus voltage control loop. The function of the voltage control loop is to maintain the DC bus voltage $V_{dc}$ as its reference value $V_{dcref}$ set by the manufacturer. When an IEMI signal introduces a deviation in $V_{a}$ on the measured bus voltage, it will lead to Equation (9):(9)$$V_{dc}^{*}=V_{dc}+V_{a},$$
where $V_{dc}^{*}$ is the DC bus voltage under attack. Then, the controller will adjust $V_{dc}^{*}$ to be equal to $V_{dcref}$, and the real DC bus voltage will become $V_{dcref}-V_{a}$ under control. This will cause the following damages.

#### 5.1.1. Breakdown of DC Bus Capacitor 

If the IEMI signal introduces a negative $V_{a}$ to the measured $V_{dc}$, the real DC bus voltage will increase and the aging of the DC bus capacitor $C_{dc}$ will accelerate. The capacitor will break down when the voltage exceeds the rated voltage of the $C_{dc}$. While the inverter incorporates over-voltage and under-voltage protection mechanisms, the vulnerability could persist, potentially leading to physical damage. This risk emerges when the adversary intentionally avoids injecting $V_{a}$ with a substantial magnitude in a single instance. This is attributed to continuously manipulating sensor values to appear within their normal range while the real DC bus voltage is spoofed. The adversary may want to ensure that during the injection of the IEMI signal, the sensor value does not trigger the under-voltage protection mechanism, allowing the IEMI to circumvent the protective measures. Afterward, the inverter loses its ability to operate correctly due to the deficiency in the $C_{dc}$’s capacity to balance the input and output power.

The simulation results are given in Figure 12a. It can be observed that the real DC bus voltage is increased by 50V, 100V, 200V and 300V after sensor manipulation. Looking at Figure 12a for the case of $V_{a}=-300\;V$, the transient voltage offset ΔV will trigger the protection instantly and shut down the inverter.

#### 5.1.2. DC Bus Under-Voltage

Similarly, an adversary can decrease the real DC bus voltage by injecting a positive $V_{a}$ into the voltage measurement. If the real DC bus voltage drops below the lower threshold, the output AC voltage will be lower than the grid voltage. In that case, the current will be reversed, and the power will flow back from the grid to the inverter, and the protection mechanisms will be triggered to shut down the inverter. This process is shown in Figure 12b when $V_{a}=100\;V$.

Hence, in summary, the impact of sensor spoofing on the DC bus voltage can be articulated as follows:

Impact 1: DoS. The DoS stops the PV inverter’s normal operation. The key of DoS is to trigger the self-protection mechanism of PV inverters. As previously analyzed, there exist two methods to induce DoS. Here, we illustrate the process by taking the example of injecting a positive deviation ($V_{a}>0$) on the DC bus voltage sensor. To achieve this objective, the adversary could design the IEMI by the following steps:

To begin, it is imperative to carefully select the frequency $f_{c+}$ of the IEMI signal through preliminary frequency testing. This choice can potentially augment the measured $V_{dc}$. Given that PV inverters of similar application levels, such as residential PV inverters ranging from 1 kW to 60 kW, typically share similar PCB dimensions, the frequencies susceptible to IEMI do not show substantial variations. Drawing from our empirical observations, $f_{c+}$ commonly falls within the range of 700MHz to 1500MHz. Subsequently, as the adversary approaches the PV inverter, it becomes necessary to transmit the IEMI signal at the designated frequency $f_{c+}$ for a brief duration, typically spanning a few seconds.

Impact 2: Damage. Damage can potentially result in the permanent breakdown of the DC bus capacitor and inflict harm upon the PV inverter. To effectuate Damage, an adversary must elevate the real $V_{dc}$ by introducing a negative $V_{a}$ into the measured $V_{dc}$ while circumventing the activation of the self-protection mechanism.

First, the adversary needs to find the frequency $f_{c-}$ that can efficiently decrease the measurement of $V_{dc}$ and generate the carrier signal $s_{c}\left(t\right)$. Since the victim system takes time to reach the stability of $V_{dc}$ after each manipulation, the adversary can design $s_{m}\left(t\right)$ as, Equation (10), where *k* and $s_{0}$ are the scale factor and initial value of $s_{m}\left(t\right)$. Generally, the smaller *k* is, the easier it is to avoid triggering the self-protection mechanism, but it takes a longer time. Finally, the adversary obtains s(t) by AM, as shown in Figure 13b.(10)$$s_{m}\left(t\right)=kt+s_{0},\;k>0,\;s_{0}\geq 0$$

To avoid triggering the protection mechanism, for the TI C2000 PV inverter [67], the target $V_{dc}$ is 385V, and the safety range is 220V∼395V. It indicates that the adversary needs to allow time for the controller to adjust $V_{dc}$ within this range after each manipulation.

### 5.2. Impact of Grid Voltage and Current Sensors

The measured grid voltage and current serve as feedback for the current control loop. Manipulations on these sensors have different effects on single-phase and three-phase PV inverters. The three-phase inverter supplies a three-phase AC power output; the phases are 120° between each other, and commonly used in industrial and commercial settings. The single-phase inverter outputs one-phase AC power, typically employed in residential PV generations.

#### 5.2.1. Single-Phase PV Inverter

We take the manipulation of grid current as an instance. If the injected deviation $I_{a}$ is constant, there will be a “transient effect” on the real grid current. This is similar to the case in which the inverter suffers from sudden grid current changes while the control loops manage to restore the current. To illustrate, let $I_{a}$ be constant and positive, then the controller will decrease the current, and the inverter’s output power will decrease. However, when the output power becomes less than the input power, the DC bus capacitor will charge, leading to $V_{dc}>V_{dcref}$, and the current reference will increase. In this regard, the reference will rise again to catch up with the manipulated current.

If the injected deviation $I_{a}$ is time-varying, like a sinusoidal signal, the PV inverter will not enter into a steady state. The simulation result is shown in Figure 14a. The larger the magnitude of the injected deviation $I_{a}$, the higher the degree of oscillation in the grid current. When the oscillation reaches a certain level, the grid current and voltage will exceed the threshold and trigger the protection mechanism, and the inverter will shut down.

#### 5.2.2. Three-Phase PV Inverter

As mentioned in the background, the three-phase voltage and current output of the PV inverter need to be transformed into the coordinate system through the Clark transformation and Park transformation before entering the control loop. In fact, due to this coordinate system transformation, a constant injected deviation into the three-phase voltage and current measurements could not affect the inverter’s output. This is because it will be filtered out by the Clark transformation matrix. Thus the Hallspoofing attacks in [12] may fail in such a scenario.

Therefore, the impact of grid voltage and current sensor manipulation in the three-phase PV inverter will only manifest when the injections are “unequal”. As illustrated in Figure 14b, compared to the single-phase inverter that needs to inject a time-varying $I_{a}$, the three-phase inverter only needs to inject a constant $I_{a}$ into one phase but not other phases to achieve a similar impact (inverter shutting down). The coordinate system is time-varying, making the component on each axis of the time-invariant signal also time-varying. We now summarize the impact of grid voltage and current sensor spoofing on PV inverters:

Impact: DoS. For DoS impact on the grid AC side, the primary adversarial strategy involves inducing oscillations in the AC voltage or current. Taking the AC current as an example, the adversary needs to inject a time-varying signal $I_{a}\left(t\right)$ on the measured AC current. We select $I_{a}\left(t\right)$ as a sine wave with the same frequency as the AC, which is not the only option.(11)$$I_{a}\left(t\right)=A_{a}\cdot sin\left(2\pi f_{AC}t\right)$$
where $f_{AC}$ is the AC frequency, and $A_{a}$ is the amplitude of $I_{a}\left(t\right)$. Since the grid imposes strict limitations on input voltage and current, an $I_{a}\left(t\right)$ with a few amps is enough to achieve the impact of DoS.

First, the adversary needs to find the frequency $f_{c+}$ and $f_{c-}$ that can increase and decrease the measured AC current. Then, they may design the modulation signal $s_{m}\left(t\right)$ as follows:(12)$$s_{m}\left(t\right)=sin\left(2\pi f_{AC}t\right)$$

Finally, obtain the IEMI signal s(t), as shown in Figure 13a,(13)$$\begin{array}{c} s\left(t\right)=\left\{\begin{array}{c} A_{+}\left(1+s_{m}\left(t\right)\right)cos2\pi f_{c+},s_{m}\left(t\right)>0,\\ A_{-}\left(1+s_{m}\left(t\right)\right)cos2\pi f_{c-},s_{m}\left(t\right)\leq 0 \end{array}\right. \end{array}$$

The adversary only needs to continuously transmit the signal for a few seconds when passing by the target inverter.

### 5.3. Impact of PV Voltage and Current Sensors

The PV voltage and current sensors are used for the MPPT algorithm and the DC–DC stage. Since the MPPT algorithm regulates the input power of the inverter by controlling the input voltage, manipulating $V_{pv}$ and $I_{pv}$ can impact the output power of the PV inverter.

Injecting a constant offset ΔV on the PV voltage sensor or ΔI on the PV current sensor only shifts the V-I curve without changing its “shape”. Thus, the MPPT algorithm will still find the correct MPP with false measured $V_{pv}$ or $I_{pv}$ by the P&O algorithm.

However, if the adversary can design a fake V-I curve with a different shape from the original one, the MPPT algorithm will be misled into finding the fake MPP, resulting in decreased power. To inject a fake V-I curve, the adversary needs to make the spoofed points ($V_{pv}$, $I_{pv}$) move on a fixed but false curve by manipulating the measured $I_{pv}$ or $V_{pv}$. We will specify this method in the following:

Impact: Damping. Damping will adversely impact the efficiency and reduce the output power of PV inverters. The primary objective of the Damping is to deceive the MPPT algorithm, preventing it from accurately identifying the MPP. Two distinct IEMI design strategies for achieving this objective exist, categorized as “spoofing” and “interference”. The “spoofing”-based method quantitatively diminishes the power output of the target PV inverter but necessitates the utilization of feedback information, namely $V_{pv}$ and $I_{pv}$ values from the internal sensors of the PV inverter. Conversely, the “interference”-based method can relatively reduce the power of the PV inverter without requiring any feedback information.

For the Damping based on “interference”: Since the MPPT finds the MPP by the P&Q method that relies on stable $V_{pv}$ and $I_{pv}$, the adversary could tamper with $V_{pv}$ or $I_{pv}$ to interfere with the MPPT. The IEMI threat can be designed akin to the DoS scenario to disrupt the measurement of $V_{pv}$ or $I_{pv}$, thereby impeding the MPPT algorithm from achieving maximum power. According to our experiment on the TI C2000 PV inverter, the injected $V_{a}$ should be between −5V and +5V to avoid triggering DoS impact instead; this threshold can be obtained by pre-test.

It is important to note that most MPPT algorithms, such as Perturb and Observe (P&O) and Incremental Conductance (IncCond), are typically designed as closed-loop control systems. The input of the closed-loop control system is the variable used to adjust the operating point of the photovoltaic (PV) array, such as the duty cycle *D*, and the output is the optimized target variable, such as the PV power $P_{pv}$, voltage $V_{pv}$, or current $I_{pv}$. As a result, there is a pole that determines the dynamic response and stability, and the imaginary part of the pole determines the oscillation frequency of the system response. Therefore, if the attacker can find the pole of the MPPT and inject a disturbance at the resonance frequency, it could potentially trigger significant instability or large-scale loss of control in the MPPT system. We assume that the closed-loop transfer function of the system T(s) can be expressed as Equations (Equation 14) and (Equation 15),(14)$$T\left(s\right)=\frac{G(s)H(s)}{1+G(s)H(s)},$$(15)$$G\left(s\right)=G_{pv}\left(s\right)\cdot G_{dc}\left(s\right)\cdot G_{controller}\left(s\right),$$
where $G_{pv}\left(s\right)$ is the dynamic model of the PV array, $G_{dc}\left(s\right)$ is the dynamic model of DC–DC converter, $G_{controller}\left(s\right)$ is the dynamic model of the controller, and H(s) is the feedback of PV power, respectively. Then in order to find the oscillation frequency of the MPPT system, we need to find the poles of the closed-loop transfer function, that is, the s-values that make the denominator zero, as shown in Equation (16).(16)1+G(s)H(s)=0.

The oscillation frequency of the MPPT closed-loop system is determined by the closed-loop poles of the system. We assume that the calculated pole is *s*, then we can obtain the oscillation frequency $f_{ocs}$ of the MPPT system, as shown in Equation (17).(17)$$s=\sigma \pm j\omega ,\;f_{ocs}=\frac{\omega }{2\pi }.$$

Since we assume the attacker can conduct a pre-test on the target inverter, she can identify the disturbance frequency $f_{ocs}$ by modeling analysis or the frequency-sweep test.

To investigate, we make the following simulation: we inject a disturbance of amplitude 1V, frequency 1Hz and 0.5Hz (System resonance frequency) into the MPPT system, as shown in Figure 15a,b. We can find that utilizing the poles of the closed-loop system can disturb the power to a greater extent.

## 6. Threat Evaluation

In this section, we first evaluate the IEMI threats on PV inverters and then test on a rural-scale microgrid operated in the real world to explore the impact on the grid. To our knowledge, this is the first work validating the IEMI threat on the real-world microgrid. To ensure the safety and legality of the research, we conducted all indoor experiments in an electromagnetic shielding room, and we contacted the manufacturer and local distribution grid operator about the testing details to avoid ethical problems.

### 6.1. Evaluation of PV Inverters

#### 6.1.1. Experiment Setup

As shown in Figure 16, the experimental setup comprises victim and adversary devices. The victim devices are off-the-shelf PV inverters and adversary devices are used to emit IEMI signals.

**Victim Devices.** To investigate the impact of IEMI attack on different solar inverters, we selected a TI C2000 inverter development kit designed by Texas Instruments in Boulevard Dallas [67], five single-phase commercial solar inverters [78,79,80,81], and a three-phase commercial solar inverter [82], as shown in Figure 17.

The inverters [67,78,79,80,83] are tested under laboratory conditions, and two models of inverters designed by GoodWe [81] are tested in a real-world microgrid.

Compared with commercial inverters, the TI inverter development kit has the following features: ① lower power and higher safety; ② most of the process variables can be read from the upper computer; ③ open-source control programs. In comparison, commercial PV inverters ① have better EMC countermeasures (such as special enclosures and internal filtering circuits); ② operate at higher power levels (several kWs), posing risks for conducting Damage experiments; thus we evaluate all three impacts on the C2000 solar microinverter and evaluate DoS and Damping on six commercial inverters.

Test-bed devices. To support the victim inverter’s operation, we use a programmable solar panel emulator TEWERD TPV1000 [84] to emulate solar panels and a RIGOL RP1025D high voltage differential probe [85] to acquire the real voltage. In particular, in the experiment of SMA three-phase solar inverter, we adopted the Chroma regenerative grid simulator 61809 [86] to simulate the three-phase grid and support the SMA three-phase solar inverter.

Adversary devices. The adversary devices are the same as those introduced in Section 4. They are used to generate, amplify, and emit IEMI signals. To prevent the adversary devices from causing conducted interference to the victim’s PV inverter through the public grid, we added a fourth-order low-pass filter between the adversary devices and the grid to eliminate conducted interference.

#### 6.1.2. Evaluation of DoS

We have introduced in Section 5 that DoS impact can be induced in two ways:

DoS on the DC side. Taking the TI C2000 inverter as an instance, we use a signal generator and RF amplifier to generate a signal with the frequency of 735MHz and the power of 10W, and emit it with the antenna. As the measured $V_{dc}$ has been tampered with, we use the high-voltage probe to acquire the real $V_{dc}$, as shown in Figure 18a.

As we can see, before DoS, the PV inverter works correctly, and $V_{dc}$ remains stable at around 385V. When IEMI is initiated, we gradually increase the measured $V_{dc}$ to “deceive” the controller. As we presupposed, the controller reduces the real $V_{dc}$, and finally, the inverter shuts down at 4.5s due to current back-flow caused by under-voltage. The process can be seen in the video (https://tinyurl.com/ReThinkDemoVideos, accessed on 20 February 2025).

DoS on the AC side. We first select the frequencies 1000MHz and 1080MHz that can, respectively, increase and decrease the measured AC voltage $V_{abc}$ through a frequency sweep. Then, we generate the IEMI signal s(t) by AM as described in Section 5. The frequency of $s_{m}\left(t\right)$ is set to be the grid frequency of 50Hz, and the total power is set to 10W, although the selection of $s_{m}\left(t\right)$ is not unique. We can see that the “Over-Grid Voltage” alarm is triggered when the measured $V_{abc}$ increases to 240V, and the “Under-Grid Voltage” alarm is triggered when the measured $V_{abc}$ is lower than 200V (https://tinyurl.com/ReThinkDemoVideos, accessed on 20 February 2025).

It is worth noting that when launching a DoS attack on the AC side of the SMA three-phase solar inverter, we do not need to inject a changing waveform into the voltage or current sensors, but only need to inject a constant deviation into one phase of the voltage or current, which can cause an imbalance in the three phases and trigger the inverter to shut down. Therefore, we only need to use a single frequency signal with constant amplitude to achieve the DoS attack on the three-phase solar inverters. The result of the DoS attack on the AC side is shown in Table 2.

#### 6.1.3. Evaluation of Damage

Damage can cause physical damage to the PV inverter by increasing the real $V_{dc}$. Through pre-test, we find that the 1350MHz IEMI signal can reduce the measured $V_{dc}$. We adjust the total power from 5W to 20W and emit it with an antenna. We use the high-voltage probe to measure the real $V_{dc}$.

The result is depicted in Figure 18b. In phase ①, the PV inverter works correctly and $V_{dc}$ remains stable at the target value of around 385V. In phase ②, we emit an IEMI signal s(t) and the controller increases the real $V_{dc}$ beyond 500V. At around 3.5s, the DC capacitor gets a dielectric breakdown and burns out after a few seconds. However, the PV inverter is “unconscious”, and $V_{dc}$ continues to rise from 3.5s to 4s. To prevent any danger, we terminate the test and cut off the power supply at 4s and the voltage $V_{dc}$ decreases to 0, as shown in video (https://tinyurl.com/ReThinkDemoVideos, accessed on 20 February 2025).

#### 6.1.4. Evaluation of Damping

Based on the analysis in Section 5, if the adversary is assumed to have feedback information such as the input voltage $V_{pv}$ and current $I_{pv}$, they can pose a greater threat by decreasing the maximum power quantitatively. Here, we focus on the scenario where no feedback information is available and evaluate the Damping impact based on the “interference” method.

For the C2000 PV inverter, we set the input power of the inverter to 80W. We first find the IEMI frequency of 1350MHz that can increase the voltage sensor’s output, and then we use the AM method to modulate the attack signal. Notably, based on the previous analysis, we carefully find the oscillation frequency $f_{oscillation}$ of the MPPT system to set as the baseband signal’s frequency by testing in the low-frequency range (e.g., 0∼100Hz). During Damping attack, we find that the inverter’s power can be reduced to 25W and cannot be automatically adjusted to 80W during the Damping. This indicates that Damping can interfere with the MPPT algorithm and reduce the inverter’s power by 68.75%.

For commercial inverters, we set the same V-I curve with a maximum power point of 2000W in the PV emulator. In the usual case, they can work stably at 1980W, 1995W and 1960W. Then, we conduct the Damping with a total power of 20W and record the power according to the PV emulator. As shown in Table 2, the power of Ginlong, Kstar, and Huawei PV inverters can be reduced by 590W, 435W and 540W at most, respectively. Similarly, the SMA three-phase solar inverter’s power can be reduced from 6000W to 4600W. Besides, we implemented the same experiment on the GoodWe inverter [81] under a real-world microgrid, and its power is reduced from 35.6kW to 2kW. The difference in reducible power is mainly caused by the perturbation resistance of different MPPT algorithms and the difference between the PV emulator in the laboratory and the real PV panel in the real-world microgrid.

Compared with DoS, Damping can be more insidious in some sense. On the one hand, it can be utilized to affect the power conversion efficiency of PV generation in the long term; on the other hand, it can launch in an on/off pattern (i.e., switching attacks) to affect the PV microgrid, as discussed in Section 7.3.

### 6.2. Evaluation of PV Microgrid

To demonstrate the threat of IEMI to the real-world grid, we collaborate with the local distribution grid operator and conduct the DoS and Damping experiments on a real-world microgrid, ensuring safety and minimal disruption to residents’ daily lives.

The microgrid has a capacity of 400kVA, and the maximum generated power of PV is 323kW. The total load is usually between 12kW and 40kW. To ensure a continuous and stable power supply, the microgrid is designed with a 150kWh battery energy storage (BES) system. It can operate in grid-connected or islanding mode, serving a discrete footprint of a remote mountain village. The PV microgrid contains two types of five PV inverters designed by GooDWe with the power of 50kW and 60kW.

In the islanding mode of the microgrid, we first evaluated DoS and Damping on each inverter. Then, we perform DoS on all five PV inverters which lasts for around 1min. We investigated the impact of the DoS on the islanding mode microgrid and recorded the frequency of the microgrid in Figure 19.

It can be observed that there is a decrease in the microgrid frequency by 1.5Hz. This shift is caused by the deficiency in PV generation at the point, prompting the BES system from the P/Q control [87] to V/f control [88]. The P/Q mode controls the output power of the PV-BES system, while the V/f mode controls the output voltage/frequency by the BES output. This indicates that the microgrid is now solely powered by the BES system, and the battery energy is continuously depleting. Notably, such a condition, mainly when the battery is low on energy, may cause more severe consequences.

However, we are not permitted to conduct the experiments under conditions of extreme low power storage that leads to over-discharging, as it could harm the health of the BES. Thus, we modeled the entire microgrid and simulated the consequence of DoS under insufficient energy storage in the simulator PowerWorld. As shown in Figure 19, the battery in the BES system is depleted in the absence of PV input for a while, and the frequency of the microgrid decreases rapidly, leading to a power outage (according to the IEEE Std 1547-2003 [89], in microgrids, the frequency deviation should not be greater than 5% of nominal). Note that as long as the PV output power is less than the load power, the BES system will continue to discharge, ultimately leading to a power outage of the microgrid.

### 6.3. Influence Quantification

Based on the principle of EMI, the IEMI distance and power can influence the threat. In this subsection, we analyze the influence of IEMI distance and power on the threat effect under the threat model.

#### 6.3.1. Influence of IEMI Distance and Power on Inverter Sensors

Here, we evaluate the effects on the deviation of the DC bus voltage $\Delta V_{dc}$ at 0∼215cm, using 5W, 10W, 20W and 50W as the total power. The result is depicted in Figure 20a. We can see that higher power allows for a greater working distance. Taking the C2000 PV inverter as an instance, the self-protection mechanism will be triggered when the $V_{dc}$ suddenly changes by 30V. With a 20W IEMI device, the inverter can be affected at a distance of around 150cm.

We placed the antenna at distances of 50cm and 100cm from the target PV inverter and tested the effects of power on the deviation of the DC bus voltage $\Delta V_{dc}$. The result is shown in Figure 20b. For the adversary’s target to generate a 30V offset on $\Delta V_{dc}$, when the distance is 50cm, the adversary only needs an IEMI power of 5W.

#### 6.3.2. Influence of IEMI Distance and Power to DoS the Commercial Inverter

Since commercial inverters respond similarly to EMI, we chose a well-selling commercial inverter, Kstar BluE-G, and recorded the maximum distance to perform DoS at a specific power. As shown in Figure 20c, we can see that a 20W IEMI can achieve DoS at a distance of 160cm, consistent with our threat model.

## 7. Discussion

In this section, we analyze the limits, diversity, and countermeasures of the proposed IEMI threats.

### 7.1. Limitation

#### 7.1.1. Subject to Power and Distance

The IEMI power and distance are crucial impact factors of the IEMI threats. Essentially, our work represents one type of attack exploiting analog signals. Such analog attacks have to follow the law of physics and a larger impact distance requires a more powerful transmitter. Notably, we find that DoS has great upward compatibility with power. For example, if a 10W IEMI signal at 50cm can shut down the inverter, then IEMI signals with 20W, 30W or even 50W can achieve the same effect. The adversary shall choose the highest possible power for success. For exploitability, attackers can disguise themselves as a passerby or remotely control drones carrying our designed portable devices, as demonstrated in video (https://tinyurl.com/ReThinkDemoVideos, accessed on 20 February 2025).

#### 7.1.2. Limited Impact Scale

Different from cyber-attacks that may cause large-scale outages, the impact of our attack is limited to PV inverters and potentially local PV microgrids. For a larger-scale grid, there may be greater resilience to compensate for the PV power. Thus, for attackers with different goals, IEMI may not always be the best approach. Besides, attackers with physical access to the inverter may launch simpler attacks with more predictable consequences. Nonetheless, IEMI attacks can be stealthier than cyberattacks in terms of digital traces, and they are also safer for attackers compared with direct physical attacks. We believe the proposed IEMI threat is applicable to local microgrid-scale attack scenarios where the attack needs to be stealthy and difficult to trace back.

### 7.2. Diversity

#### 7.2.1. Diversity of the Impact

We propose DoS, Damage, and Damping to illustrate the threat of IEMI. Since IEMI can control multiple sensors simultaneously, adversaries can use it to explore more impacts, such as controlling the output frequency, the output power factor, and more. For example, IEMI can also introduce harmonics (using the method in Figure 11) into the AC output of the inverter and damage electrical appliances or devices.

#### 7.2.2. Diversity of the Victim

This study highlights the vulnerability of op−amp−based voltage and current sensors in PV inverters to EMI. While a PV inverter is a typical example of a power electronic device, the scope of potential victims can extend. Similar sensor technologies and energy conversion processes are prevalent in various applications, including power grids, electric vehicles, and industrial machinery. Additionally, the control algorithms employed in different inverters partly exhibit similar characteristics. For instance, the battery storage inverter may adopt the TSPC system [90], implying the presence of a DC bus capacitor in such inverters and the associated impact of Damage and DoS. Consequently, it is imperative that the security analysis should also be performed in these diverse domains.

### 7.3. Exploitability

#### 7.3.1. Large-Scale Impact

The proposed IEMI impacts may cause consequences to the microgrid that go beyond those achieved in our evaluation, under specific conditions where there are both solar PV and synchronous generators in a grid. Particularly, for the Damping that can manipulate the output power of the PV inverter by more than 90% (as tested in the real-world microgrid), it can launch in an on/off pattern and induce low-frequency oscillations of power supplies, which may cause physical damage of other synchronous generators and even result in a power outage, similar to how Switching Attacks [91] affect the grids [92]. This is because, the low-frequency oscillations can result in angular speed oscillations of generators, which can lead to damage or disconnecting of the generators. It has been demonstrated that manipulating a mere 1.23% of the total system power is enough to achieve the Switching Attack [27]. To further verify, we simulate the use of Damping to oscillate the angular velocity of generators in the grid (the modified Kundur benchmark system with four synchronous generators and two PV farms [93]) via Simulink, and our simulation result shows that Damping could cause this cascading failure effectively, as shown in Figure 21.

#### 7.3.2. Closed-Loop Attack

One limitation of this work is that the attacker has no access to the sensor’s output as feedback to launch closed-loop attacks, such as quantitatively decreasing the output power. Notably, side-channel attacks can exploit physical side effects such as power consumption, electromagnetic emissions, or even timing variations in the system to extract sensitive information, such as the voltage and current value. Therefore, this work can be further improved by combining it with side-channel attacks in the future.

### 7.4. Electromagnetic Compatibility Standards

Grid operating parameters and Electromagnetic Compatibility (EMC) standards for photovoltaic (PV) inverters vary significantly across regions. For instance, SMA PV inverters designed for the European market must comply with the EN 61000 series standards [94], while those tailored for other markets adhere to the GB/T 17626.x standards [95]. They have different requirements in terms of scope of application, test methods, test limits, and so on.

However, we find that all tested solar inverters conforming to different EMC standards remain vulnerable to certain IEMI attacks. We believe this vulnerability primarily arises from the non-ideal characteristics of electronic components in EMC designs. For example, the differential op amp circuit cannot completely eliminate common mode interference, and the filter device has filter leakage for high-frequency noise. The existence of these physical hardware vulnerabilities makes it difficult to completely eliminate the IEMI threats improving EMC standards. Therefore, we prefer to look into proactive detection defense methods to deal with this type of threat.

## 8. Countermeasures

Since the sensor’s deviation under the IEMI attack is similar to the normal operating conditions, current IEMI attack detection methods cannot determine the reliability of the sensor data. In this section, we investigate three methods from the signal level, model level, and combination level, respectively, aiming at converting serious threats (e.g., physical damage) into light threats (e.g., DoS) and providing timely alerts to managers.

### 8.1. Detection on the Sensor Level

The coupling of IEMI has the distributed effect [96], which means that IEMI cannot be injected into the target node individually but will affect multiple nodes at the same time. During the attack, the wanted IEMI noise is injected into the input node of the voltage sensor. It is rectified, amplified, and filtered by the op−amp circuit, resulting in a DC deviation. At the same time, the IEMI will also induce other unavoidable effects, which can be leveraged as detection features. For instance, the output node of the voltage sensor can also cause noise and superimpose to the sensor deviation, as shown in Figure 22.

To investigate, we conducted an IEMI attack experiment on the Ti C2000 solar inverter. We recorded the DC bus voltage sensor’s output at a sample rate of 1kHz. To examine the impact of IEMI frequency, we employed the IEMI with the frequency of 1604MHz, 1236MHz and 1560MHz to increase the sensor value, and used the IEMI with the frequency of 1740MHz and IEMI with the frequency of 1726MHz to decrease the sensor value, maintaining the same IEMI power and distance (7W) and distance (10cm).

The sensor’s output is shown in Figure 23. We can see that different frequencies can cause different deviations, but the IEMI noise on the sensor’s output is not significant. This is mainly because the IEMI does not form a resonant electromagnetic coupling to the sensor’s output node.

Considering that the inverter sensor’s sample rate (1kHz) is much lower than the IEMI frequency, IEMI noise cannot be clearly distinguished from normal noise in terms of frequency, but the sample rate will not limit the noise amplitude. Thus, we select the Standard Deviation (STD) as the feature of the IEMI noise, quantifying the variation or dispersion in a set of data values and indicating how much the data points deviate from the average. The STD can be expressed as Equation (18):(18)$$s=\sqrt{\frac{1}{n-1}\sum_{i=1}^{n}{\left(x_{i}-\bar{x}\right)}^{2}}.$$

Then, we recorded the sensor’s output and calculated the STD under normal conditions and during IEMI attacks of different frequencies. The sensor’s output is depicted in Figure 23, and the STD is illustrated in Figure 24. We observed that (1) IEMI of the same power but different frequencies leads to different STDs on the sensor output. This is mainly because IEMI of different frequencies causes different coupling efficiencies and further induces different noise at the sensor output node; (2) regardless of whether IEMI increases or decreases the sensor output, the STD exceeds that under normal conditions. Therefore, we can conclude that the STD can serve as a feature to detect IEMI attacks.

Impact of attack power. According to our above analysis, a higher IEMI power may produce a larger STD to the output of the sensor output. To investigate, we attacked the sensor using IEMI with different power (4.47W, 5.01W, 5.62W, 6.31W and 7.08W, respectively) and the same frequency (1560MHz). The sensor’s output under different IEMI powers is shown in Figure 25, and the sensor’s STD is shown in Figure 26. We can find that higher power can cause a larger STD of the sensor’s output.

### 8.2. Detection on the Model Level

#### 8.2.1. Detection Principle

Since a PV inverter is an energy converter, it does not produce or consume energy by nature (except for a small amount of loss). At the same time, the sensor’s value may violate this physical law when manipulated by IEMI attacks. Therefore, we propose a detection algorithm based on the inverter’s energy conservation law.

In general, the difference between the input power and output power of the inverter in a steady state represents the circuit’s loss power, which can be expressed by Equation (19):(19)$$I_{pv}V_{pv}-I_{ac}V_{ac}\cos\phi -P_{\delta }=0,$$
where the $I_{pv}$ and $V_{pv}$ are the input current and voltage of the PV panel, $I_{ac}$ and $V_{ac}$ are the output current and voltage to the grid, cosϕ is the power factor (generally 0.95∼1), and $P_{\delta }$ is the power losses due to transformers and switch devices inside the inverter (generally accounts for about 8∼13%). To simplify, we can calculate the energy conversion efficiency σ as Equation (20):(20)$$\sigma =\frac{I_{ac}V_{ac}}{I_{pv}V_{pv}}.$$

Since many International standards [97,98,99] require a conversion efficiency of at least 90% for the PV inverter’s regular operation, the σ should be around 0.9 during regular operation. If σ<0.8 or σ>1, it indicates that at least one of $I_{pv}$, $V_{pv}$, $I_{ac}$, or $V_{ac}$ has been manipulated, such as in the Damping attack. For attackers, it is challenging to control $I_{ac}$ and $V_{ac}$ in real-time to maintain a constant value, making it difficult to bypass the detection.

#### 8.2.2. Evaluation

To investigate whether the proposed detection method, based on inverter features, can detect IEMI attacks and distinguish them from the power degradation caused by natural factors (such as temperature drop and cloud cover), we designed and implemented the following experiment:

The proposed IEMI attacks involve $I_{pv}$, $V_{pv}$, $I_{ac}$ and $V_{ac}$ and include Damping and DoS attacks; among them, the Damping attack covertly and continuously interferes with the MPPT algorithm of the PV inverter to reduce the power. Here, we take the covert Damping attack as an example to evaluate the detection effect. We used the programmable solar panel emulator TEWERD TPV1000 [84] to emulate solar panels, and used a Ti C2000 micro solar inverter as the target inverter to record the $I_{pv}$, $V_{pv}$, $I_{ac}$, and $V_{ac}$ at a sample rate of 1kHz. We first made the inverter work at 100W, and then implemented the Damping attack based on “interference”. After the attack, we reduced the input power by half to simulate the environmental disturbance and investigate whether this detection method would misjudge a regular environmental disturbance as an attack.

The result is shown in Figure 27. As we can see (1) from 0.2s to 0.8s, the input and output power has a fluctuation within 10V, and the σ oscillates between 0.9 and 1; (2) from 0.8s to 3s, the PV inverter works in a normal state and the σ is stable at 0.95; (3) from 3s to 3.2s, we implement the Damping attack based on interference and the σ oscillates more than 0.2, the Damping attack is detected; finally, from 4s to 5s, we reduced the input power by half to simulate the environmental disturbance, we can see that although the input and output power reduced, the σ is still stable between 0.9 and 1, which illustrates that the detection method could bypass the regular environmental disturbance.

#### 8.2.3. Impact Factors

Although we have successfully detected a Damping attack on the C2000 solar inverter, the detection effect (σ value) may be affected by many factors, such as the inverter’s working power and the attack strength. We conducted the following experiment to explore the impact of working power and attack strength.

Impact of working power. Since each component within the inverter has different operating efficiencies at different working powers, the working power of the inverter may affect the efficiency σ. In this experiment, we set the Ti C2000 solar inverter to operate at 0∼240W, respectively, and calculate σ. The result in Figure 28a shows that the inverter’s efficiency σ can be kept above 0.9 if the working power is greater than a threshold value (such as 40W).

Impact of attack strength. Since the proposed detection methods rely on the sensors’ deviation determined by the attack strength, we need to investigate the impact of the attack strength on the detection effect. We implemented the Damping attack on the Ti C2000 solar inverter by inducing a 0∼18V deviation on the PV input voltage sensor and calculated the deviation of σ under different attack strengths.

The result shown in Figure 28b indicates that IEMI-induced sensor deviations of 4V or more can cause σ to oscillate by more than 0.2, which can be easily detected. In contrast, IEMI-induced sensor deviations of less than 4V cannot be detected and do not pose significant threats to the inverter. Therefore, we can conclude that most attacks manipulating $I_{pv}$, $V_{pv}$, $I_{ac}$, and $V_{ac}$ will cause a detectable deviation in the σ.

### 8.3. Detection on the Combination Level

Since an inverter is a relatively fixed system, there are complex intrinsic connections between the various sensors and neural networks are better at extracting these features. Based on this idea, we explore a neural network-based detection method, which is more likely to become a future direction.

Condition analysis. To build a neural network model and deploy it on the inverter’s MUC (Microcontroller Unit, e.g., Ti C2000), we need to consider the following factors:(1)Data: The input data need to take into account the intrinsic connections between different sensors, and the intrinsic connections between a sensor’s data frames;(2)Model: The training of the model before leaving the factory can be offline, but it needs to be online to detect anomalies after being deployed to the inverter, so it is important to conserve arithmetic as much as possible;(3)Deployment: Since IEMI attacks take effect in seconds, the inverter only needs to detect IEMI attacks at second-level intervals.

Dataset building. Since this paper presents a completely new threat to PV inverters, there is no open-source dataset of this threat. Here, we take the Ti C2000 micro solar inverter as an example and build the dataset by collecting five sensors’ data under normal and attack conditions. To cover different normal conditions, we set the inverter to work at 40W, 60W, 80W and 100W, respectively. To cover different attack conditions, we set the attack power to 5W, 10W, 15W, 20W, respectively. We take 100 frames of five sensors’ data of the inverter in 0.1s as a sample (5×100), and collect a total of 5000 samples of data, containing 29.5% positive samples (IEMI attack) and 70.5% negative samples (No attack). Among them, 4000 samples are set as the training set, 500 samples as the testing set, and 500 samples as the validation set. The label of each sample is denoted by 0 or 1, with 0 representing no attack and 1 representing an IEMI attack.

Model building. In order to extract the intrinsic connections between different sensors and different data frames, we employed a lightweight convolution neural network (CNN) with 5×100 matrix inputs to achieve the binary classification tasks. As shown in Figure 29, the model consists of two convolution layers, a flattened layer and a fully connected layer. The first convolution layer adopts four filters of size 2×2 with a stride of 1, followed by a max-pooling layer with a 2×2 window and a stride of 2. The second convolution layer adopts eight filters of size 2×2 with a stride of 1, followed by another max-pooling layer with the same window size and stride. The output from the second pooling layer is flattened into a 200×1 vector and passed through a fully connected layer for binary classification. The model is computationally efficient, making it well-suited for resource-constrained applications. The model contains a total of about 18,400 multiplication and addition operations, and for the Ti C2000 microcontroller, it takes about 0.46ms to complete an operation.

Evaluation. Here, we evaluate the effectiveness of CNN models in detecting IEMI attacks. The test set of 500 samples obtained from the Ti C2000 micro solar inverter contains 147 positive samples (of the IEMI attack) and 353 negative samples. We used the trained model to classify the test data, and the results showed that 344 out of 353 sets of negative samples were identified as negative samples and nine sets were incorrectly identified as positive samples; 141 out of 147 sets of positive samples were identified as positive samples, and six sets were incorrectly identified as negative samples, as shown in Table 3. To gain a more comprehensive understanding of the performance of the binary classification model, we further calculated four key metrics: Accuracy, Precision, Recall, and the F1 Score, as shown in Table 4.

### 8.4. Comparison of the Three Detection Methods

Among the three proposed detection methods, the detection based on the distributed effect of IEMI can detect attacks on any sensors, but the noise feature can be affected by attack parameters; the detection based on the conservation of energy only needs to calculate the efficiency of the inverter, but it can only detect attacks on the input and output sensors; the detection based on the neural network can extract features and detect attacks most efficiently, but brings more arithmetic expense.

The detailed comparison is shown in Figure 30. In conclusion, we believe the third method is most likely to be the future direction because the detection effect and arithmetic consumption can be significantly optimized by improving neural network models in the following work.

## 9. Conclusions

This study presents a comprehensive analysis of the security vulnerabilities in photovoltaic (PV) inverters, focusing on the effects of intentional electromagnetic interference (IEMI) signals around 1 GHz on their voltage and current sensors. Three primary impacts are identified: DoS, which causes inverter shutdowns; Damage, leading to physical component failure and Damping, which reduces power output. A thorough evaluation of seven different commercial PV inverters and a real-world microgrid demonstrates that all of these systems can be attacked by a 20 W IEMI signal at distances ranging from 1 to 1.5 m. The limitations, variability, exploitability, and root causes of these vulnerabilities are also examined. To mitigate these risks, three detection methods are proposed and assessed: sensor-level detection, model-level detection, and combination-level detection, with a detailed discussion of their advantages and limitations. In conclusion, these findings highlight the increasing security concerns surrounding power electronic devices in grids that are becoming more reliant on renewable energy sources (RES) and aim to provide ideas for the designers and manufacturers in the future.

## Figures and Tables

**Figure 1 sensors-25-01493-f001:**
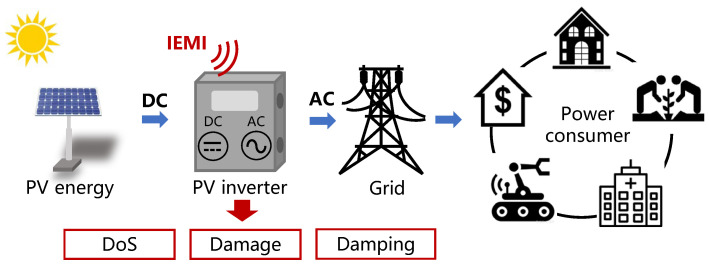
An illustration of the IEMI threat: IEMI can affect PV inverters and cause DoS or physical damage, or damping the power output.

**Figure 2 sensors-25-01493-f002:**
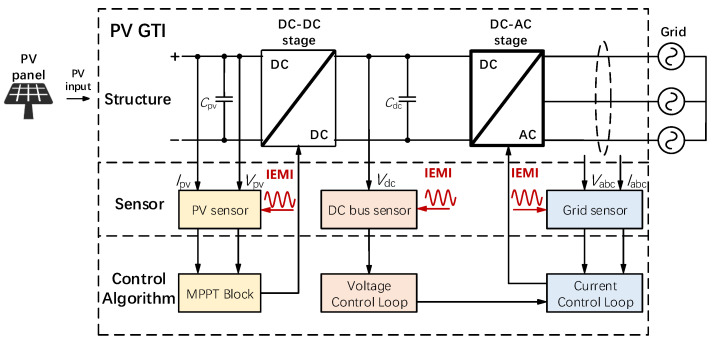
A typical PV inverter can be modeled as a three-layer structure: Power conversion unit-Sensor-Control algorithms.

**Figure 3 sensors-25-01493-f003:**
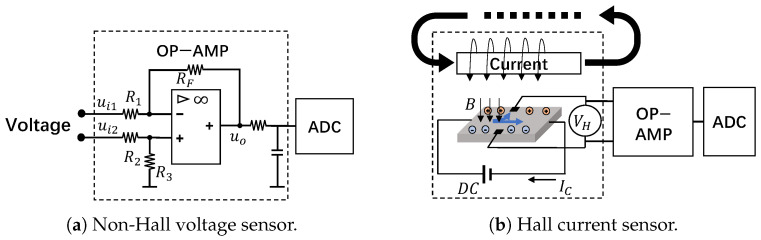
The schematic of voltage and current sensors in the PV inverter.

**Figure 4 sensors-25-01493-f004:**
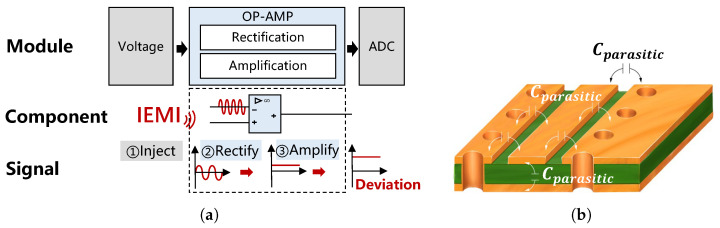
The principle of IEMI impact on voltage sensors. The IEMI signal is coupled into the sensor circuit, and then rectified, amplified by the op−amp, and ultimately turned into an offset on the output. (**a**) Transmission process of IEMI signals in the voltage sensor. (**b**) The parasitic capacitance of sensor’s PCB.

**Figure 6 sensors-25-01493-f006:**
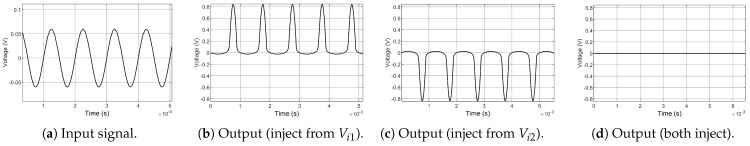
Simulation of IEMI injection on different inputs of the op−amp chip.

**Figure 7 sensors-25-01493-f007:**
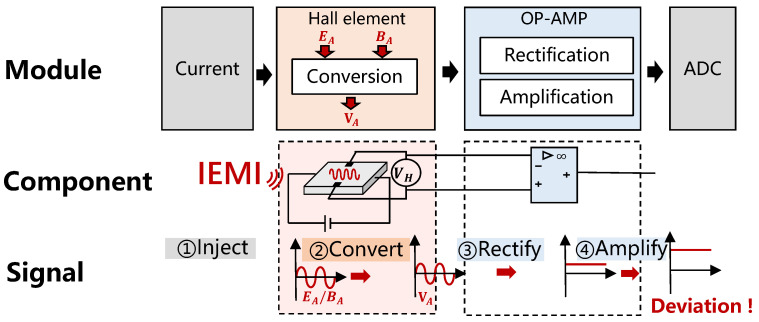
The principle of IEMI impact on Hall current sensors. The IEMI signal is injected into the Hall chip and generates a noise $V_{H}$. Then the noise will be rectified, amplified by the op−amp, and result in a deviation on the output.

**Figure 8 sensors-25-01493-f008:**
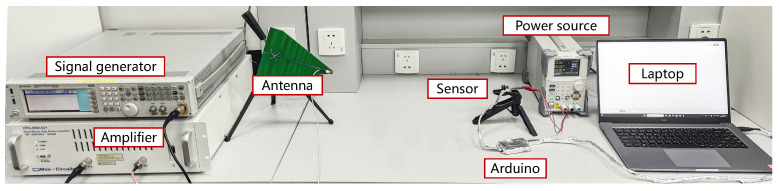
Setup of feasibility test on sensors.

**Figure 9 sensors-25-01493-f009:**
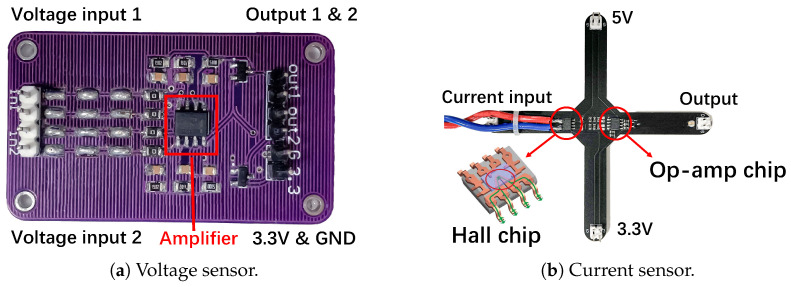
The voltage and current sensors’ PCB we designed for the initial feasibility test.

**Figure 10 sensors-25-01493-f010:**
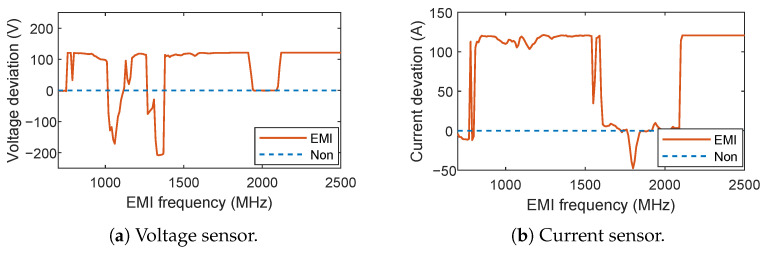
The result of the IEMI frequency test on the voltage and current sensors. The IEMI power and distance are set to 10W and 50cm.

**Figure 11 sensors-25-01493-f011:**
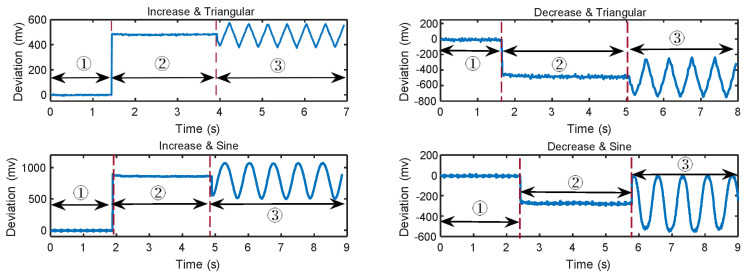
The experiment result of manipulation with a single-frequency signal and an AM signal on the sensor. ①: Without EMI; ②: Single-frequency EMI; ③: AM-modulated EMI.

**Figure 12 sensors-25-01493-f012:**
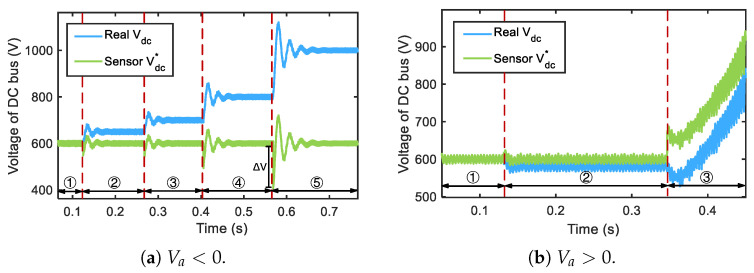
The simulation of the DC bus voltage manipulation. We add a fake $V_{a}$ on the measured DC bus voltage and record the real DC bus voltage under control. For $V_{a}<0$, ①: $V_{a}=0$ V, ②: $V_{a}=-50$ V, ③: $V_{a}=-100$ V, ④: $V_{a}=-200$ V, ⑤: $V_{a}=-300$ V; for $V_{a}>0$, ①: $V_{a}=0$ V, ②: $V_{a}=20$ V, ③: $V_{a}=100$ V.

**Figure 13 sensors-25-01493-f013:**
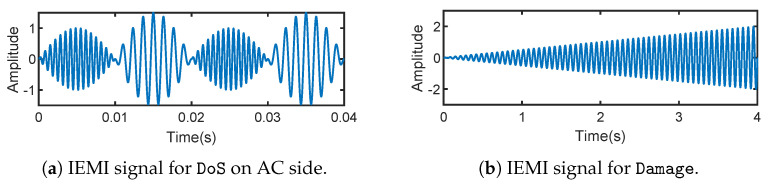
Design of IEMI signals s(t) of DoS and Damage.

**Figure 14 sensors-25-01493-f014:**
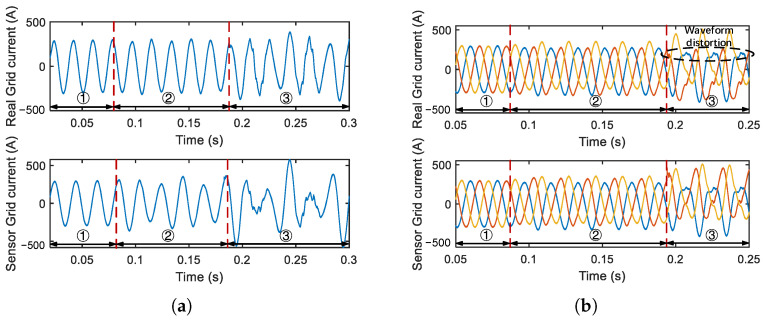
The simulations of grid current sensors spoofing. It gives the simulated waveform of the real current value and the sensor output value when the single-phase and three-phase grid current measurement is manipulated. (**a**) Single-phase PV inverter. ①: $I_{a}=0$ A, ②: $I_{a}=50\sin\omega t\;$ A, ③: $I_{a}=200\sin\omega t\;$ A. (**b**) Three-phase PV inverter. ①: $I_{a}=0$ A, ②: $I_{a}=50$ A, ③: $I_{a}=200$ A.

**Figure 15 sensors-25-01493-f015:**
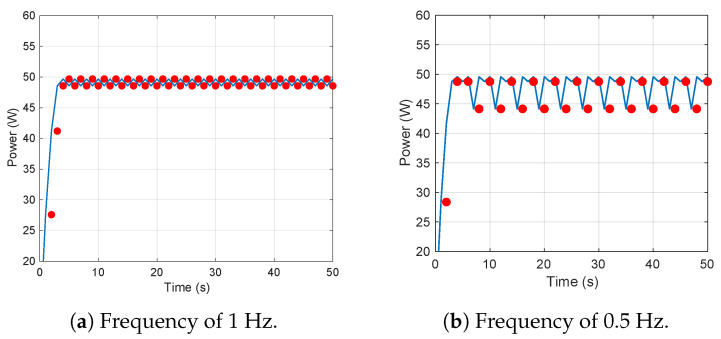
Simulation of injecting perturbations of different frequencies into the MPPT control system. The red dots represent the positions at which the perturbations are injected.

**Figure 16 sensors-25-01493-f016:**
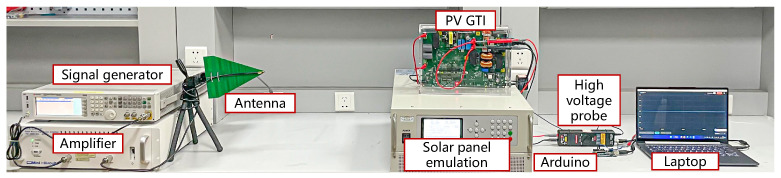
Experiment setup of evaluation on PV inverters.

**Figure 17 sensors-25-01493-f017:**
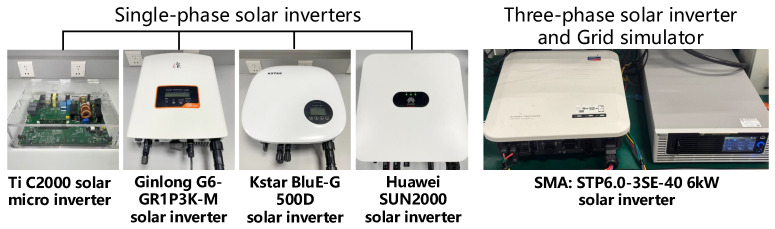
The tested single-phase solar inverters and three-phase solar inverters under laboratory conditions.

**Figure 18 sensors-25-01493-f018:**
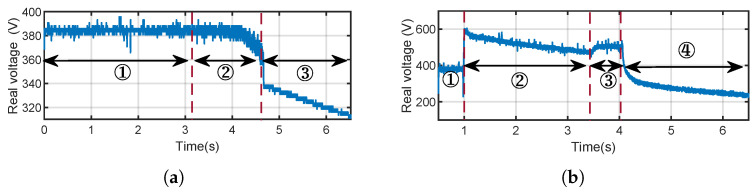
The experiment results of DoS and Damage. (**a**) Result of DoS. ①: Before EMI, ②: IEMI begins, ③: After EMI. (**b**) Result of Damage. ①: Before EMI, ②: IEMI begins, ③: Burning out, ④: After EMI.

**Figure 19 sensors-25-01493-f019:**
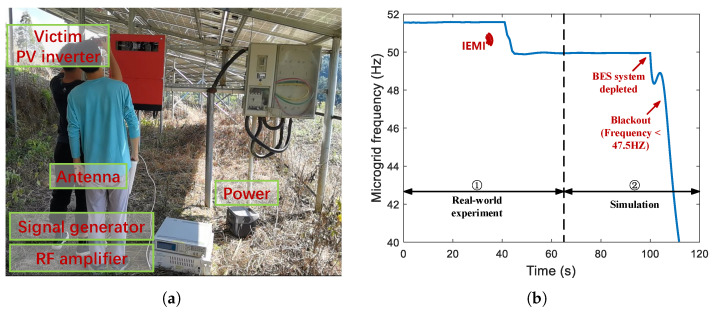
The impact of DoS on a real-world PV microgrid’s frequency. Stage ①: real-world experiment, Stage ②: simulation. (**a**) Experiment setup in the real-world microgrid. (**b**) Impact of DoS on microgrid frequency.

**Figure 20 sensors-25-01493-f020:**
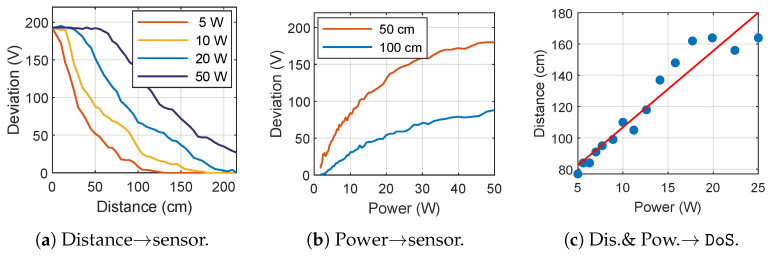
The influence of distance and power to manipulate inverter sensors and DoS a commercial inverter. The nonmonotonicity in (**c**) is mainly because the power will affect the electromagnetic field distribution of the antenna, which is not linear.

**Figure 21 sensors-25-01493-f021:**
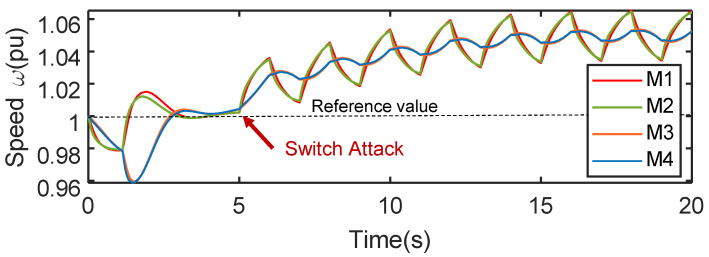
The simulation result of Switching Attack with Damping.

**Figure 22 sensors-25-01493-f022:**
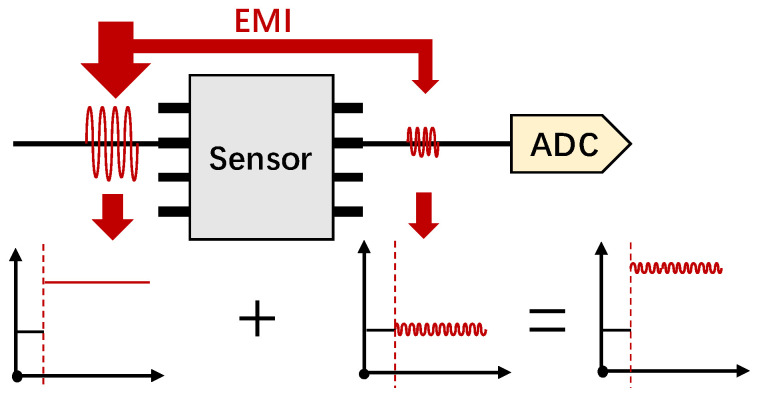
The detection method based on the distributed effect of IEMI. IEMI coupled before the transducer is converted to DC bias, while IEMI coupled behind the transducer remains AC noise, which can be regarded as a detection feature.

**Figure 23 sensors-25-01493-f023:**
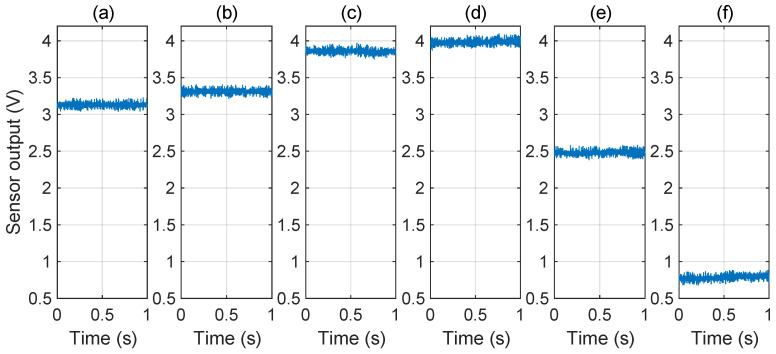
The voltage sensor’s output under different IEMI attack frequencies. (**a**) is under normal state, (**b**–**f**) are under IEMI attack with the attack power of 7W and frequency of 1604MHz, 1236MHz, 1560MHz, 1740MHz and 1726MHz.

**Figure 24 sensors-25-01493-f024:**
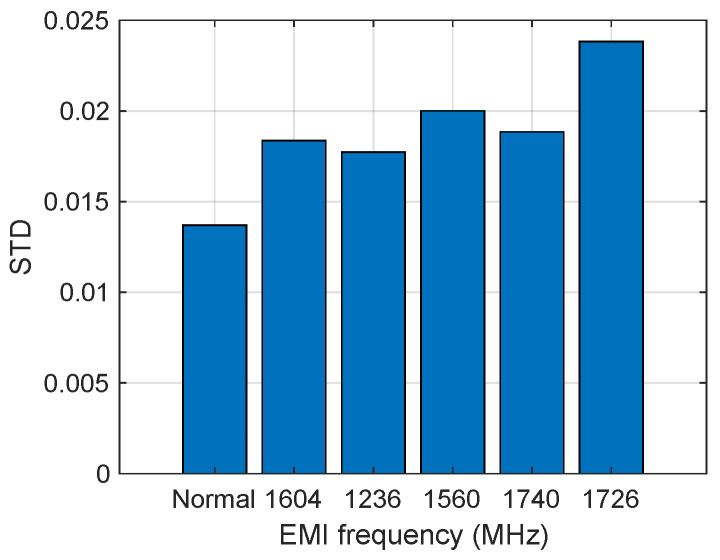
Sensor output’s STD under different IEMI frequencies.

**Figure 25 sensors-25-01493-f025:**
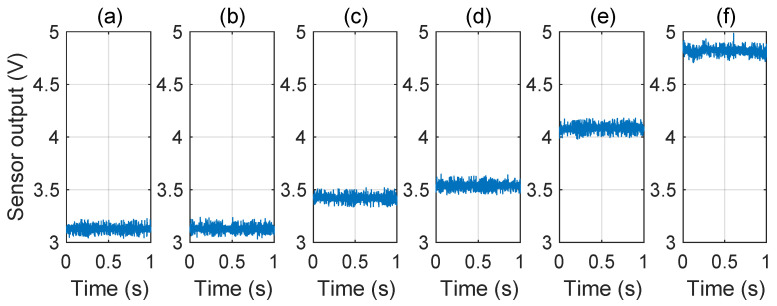
The sensor’s output under different IEMI attack power. (**a**) is under normal state, while (**b**–**f**) are under IEMI attack at a frequency of 1560MHz and power levels of 4.47W, 5.01W, 5.62W, 6.31W, and 7.08W, respectively.

**Figure 26 sensors-25-01493-f026:**
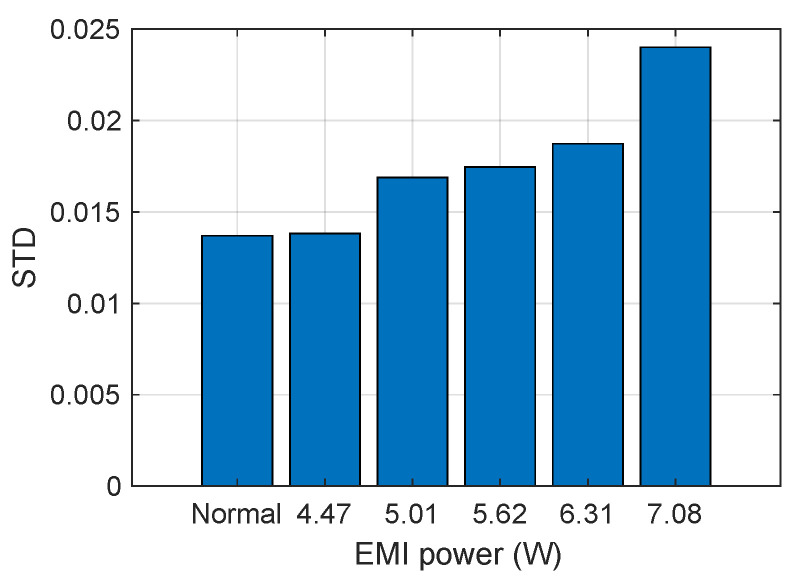
Sensor output’s STD under different IEMI power levels.

**Figure 27 sensors-25-01493-f027:**
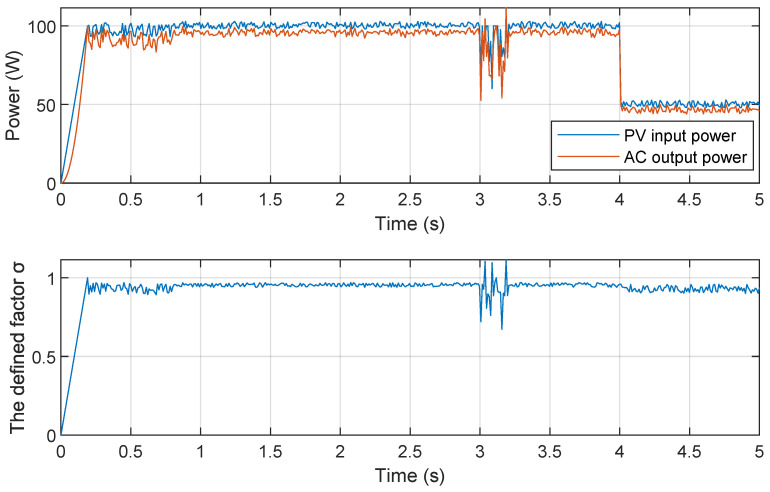
The experiment result of the detection of Damping attack on the TI C2000 solar inverter. 0∼0.8s: Initialization, 0.8∼3s and 3.2∼4s: Normal operation, 3∼3.2s: Damping attack, 4∼5s: Manual reduce power by half.

**Figure 28 sensors-25-01493-f028:**
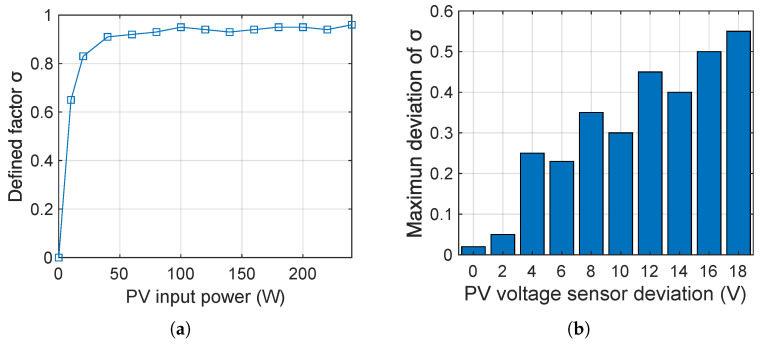
The impact of inverter working power and sensor’s deviation under attack on σ. (**a**) The efficiency σ under different working power. (**b**) The maximum deviation of efficiency σ under different sensor’s deviation caused by IEMI attack.

**Figure 29 sensors-25-01493-f029:**
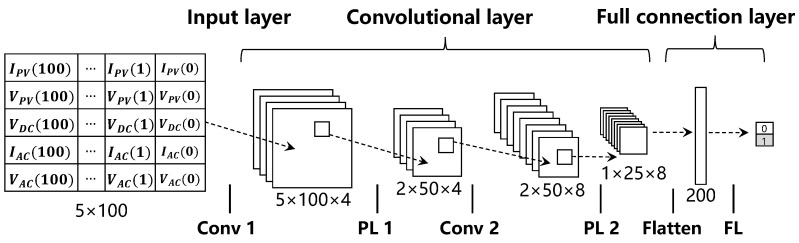
The structure of the lightweight CNN model. Including 2 convolution layers, a flattened layer and a fully connected layer.

**Figure 30 sensors-25-01493-f030:**
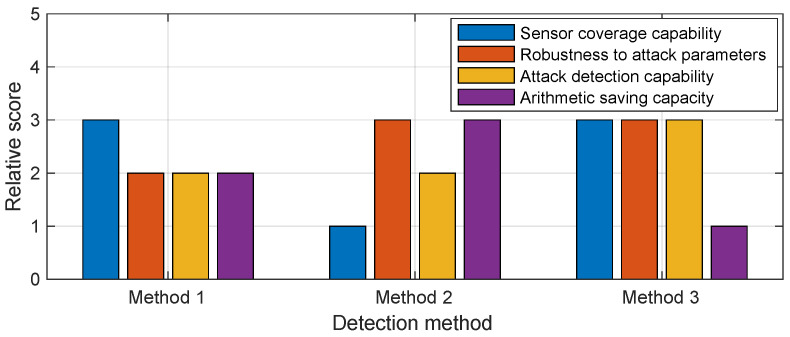
The comparison of the three methods. Method 1 is based on the distribution of IEMI, method 2 is based on the conservation of energy, and method 3 is based on neural networks. The “3” means excellent, “2” means good, “1” means fair.

**Table 1 sensors-25-01493-t001:** Result of IEMI impact on seven Hall sensors.

Sensor Type	Sensor Model	Output Type	Measure- Ment Span	Test Parameters		Output				
				Freq. (MHz) (Pos./Neg.)	Pow. (W)	Original Value	Pos. Dev.	Pos. Dev. Rate	Neg. Dev.	Neg. Dev. Rate
Current	WCS1800 (Wire)	Analog	0∼30 A	685/1030	10	5 A	15.7 A	+214.00%	−1.1 A	−1.00%
Current	WCS1800 (Wireless)	Analog	0∼35 A	1000/876	10	5 A	31.5 A	+530.00%	−1.6 A	−1.00%
Current	ACS712 (20 A)	Analog	0∼20 A	779/1223	10	5 A	13.2 A	+164.00%	−1.2 A	−1.00%
Current	ACS712 (5 A)	Analog	0∼5 A	627/1212	10	2.5 A	5.1 A	+104.00%	−1.75 A	−1.00%
Speed	3144	Digital	0/1	677	10	0/1	bit-flap	+100.00%	bit-flap	−1.00%
North pole	3144	Digital	0/1	724	10	0/1	bit-flap	+100.00%	bit-flap	−1.00%
Water flow	YF-S401	Digital	0/1	1322	10	0/1	bit-flap	+100.00%	bit-flap	−1.00%

**Table 2 sensors-25-01493-t002:** Result of IEMI attacks on PV inverters.

Inverter	DoS						Damage			Damping			
	On DC Side			On AC Side			Pow. (W)	Freq. (MHz)	Result	Freq. (MHz)	Pow.(W) Before Damping	Pow.(W) After Damping	Pow. Dev. Rate
	Pow. (W)	Freq. (MHz)	Success Rate	Pow. (W)	Freq.(MHz) Pos./Neg.	Success Rate							
Ti C2000	5	735	100%	5	1036/1490	100%	10	1000	100%	760	80	25	68.75%
Ginlong	10	916	100%	10	625/1210	80%	-	-	-	1192	1980	1390	29.8%
Kstar	10	749	100%	10	990/810	90%	-	-	-	998	1995	1560	21.8%
Huawei	10	1150	100%	10	980/1020	80%	-	-	-	1330	1960	1420	27.6%
SMA	10	675	100%	10	1125	100%	-	-	-	753	2950	2660	9.8%
GW (LCD, 50 kW)	20	920	100%	-	-	-	-	-	-	960	35.6k	2k	94.3%
GW (LED, 60 kW)	20	945	100%	-	-	-	-	-	-	-	-	-	-

**Table 3 sensors-25-01493-t003:** Confusion Matrix for Model Evaluation.

	Predicted Positive	Predicted Negative
Actual Positive	141 (TP)	6 (FN)
Actual Negative	9 (FP)	344 (TN)

**Table 4 sensors-25-01493-t004:** Evaluation Metrics.

Metric	Accuracy	Precision	Recall	F1 Score
Value	94%	96%	97%	97%

## Data Availability

The data presented in this study are available on request from the corresponding author.

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
