# Peer review of "Systematic Security Analysis of Sensors and Controls in PV Inverters: Threat Validation and Countermeasuresâ€"

_sensors, 2025, doi:10.3390/s25051493_

Round 1
Reviewer 1 Report
Comments and Suggestions for Authors
In my opinion this is an excellent paper, with a very good study of the security vulnerabilities of PV inverters, complete with models and lab and field tests. I am sure that it will set a milestone in this field of research.
The exposition is very clear and not redundant, tables and figures are accurate and well integrated in the text.
In conclusion, I think that this work is well suited to appear in Sensors in the present form.
Reviewer 2 Report
Comments and Suggestions for Authors
The paper systematically analyzes the safety of photovoltaic (PV) inverter sensors and control systems, with a focus on the impact of electromagnetic interference (EMI) on embedded current and voltage sensors. The paper introduces the ReThink attack method, demonstrating that EMI can lead to inverter denial of service (DoS), physical damage, and power damping, and provides experimental verification in six commercial single-phase and three-phase inverters as well as a real microgrid. Additionally, the paper proposes signal-level, model-level, and combined-level detection methods to address EMI threats and enhance inverter security. This study has certain theoretical value and engineering application significance, but there are still some shortcomings. It is recommended to minor revise the paper. The specific revision suggestions are as follows:
·It is suggested to check the font formatting throughout the entire paper, e.g., in Section 2.2.1 (the last paragraph), the first paragraph of Chapter 5, and the last paragraph of Section 5.1.1, where the font within the paragraphs is inconsistent.
·It is suggested to check the definition of formula symbols. Some formulas in the paper do not define symbols when they first appear. For example, some variable symbols are not clearly explained in the context, which may lead to difficulty in understanding. It is recommended to check the formulas one by one to ensure that all variables have clear mathematical or physical meaning explanations.
Comments on the Quality of English LanguageOptimize language expression to improve readability. The paper contains detailed content, but some paragraphs are somewhat lengthy. It is recommended to streamline certain expressions to make the conclusions clearer and enhance overall readability and logical coherence.
Reviewer 3 Report
Comments and Suggestions for Authors
Thanks for the submission. This work discusses the security threats and countermeasures of sensors and controls in PV inverters. It has been well-written. I only have some minor comments:
(1) PV inverters serve as an important part of modern power grid operation. As mentioned in lines 41-44, malicious EMI signals may be a concern. Actually, there is a professional definition, namely IEMI. Please refer to and discuss "A review of intentional electromagnetic interference in power electronics: Conducted and radiated susceptibility" in this part.
(2) As discussed in lines 56-70, the EMC usually can be divided into two parts: EMI and electromagnetic susceptibility. The effects of intentional (malicious) EM signals belong to the latter one. It is good to provide more clarification. Please refer to comment 1.
(3) To trick the inverter's control algorithms, you may not be able to ignore the influences of side-channel attacks.
(4) For Section 1.3, when you mentioned the filtering techniques, the key point of filtering components in defending EMI attacks is their high-frequency characterization and modeling. The latest work in review, characterization, impedance modeling, impedance measurement, and numerical study can be expanded. You may find more details in the latest review paper published in IEEE TPE (Characterization and Circuit Modeling of Electromagnetic Interference Filtering Chokes, Section III-B). I agree with you about the limitations of passive countermeasures but maybe the hybrid will be more suitable.
(5) For the sensors, usually the most critical failures come from the front-door coupling.
(6) As seen in the experimental setup, you perform the study from the individual modules but not from a whole device. Will it cause over-design or under-design due to the complex coupling path in real products? In addition, I believe the PV PCB will not be exposed to the air directly. There is the metal box. Maybe you can consider it in the future.
Reviewer 4 Report
Comments and Suggestions for Authors
Review “Systematic Security Analysis of Sensors and Controls in PV
Inverters: Threat Validation and Countermeasures” in Sensors
Thanks for offering me the possibility of developing this review.
The technical write content and the videos are relevant, but there are reference issues, content redaction problems outside the technical measurement and results, and a big problem with the inappropriate use of the authors term “ReThink”.
Even with these problems, the article could be publishable if the authors make a great effort to cut out non-relevant text, improve the writing, especially the conclusions, and solve the problem with the term ReThink.
Before line 1 I read. “.This manuscript is an expanded version of our previously published conference paper [1].” I think that an ArXiv manuscript will be a conference paper, if this is a conference peer that need to be identify in the reference.
I call the attention to the editor over the possibility that this publication appear in ArXic in another conference and in sensors.
The authors “ we focus on the distinct security of inverters, i.e., the security of analog 36
sensors, since inverters rely on correct sensing of voltage and current of input power 37
sources as well as the grids to ensure stable and safe power conversion” due ONLY to intentional electromagnetic interference (IEMI) attack, but this second part it is not explicitly write in the manuscript, this suppose that for several pages the reader must find clues (such as videos) and play detective to understand what specific topic the manuscript is investigating. This together with an insufficient explanation of the term “ReThink” it makes it difficult to clearly understand the objective of the article.
I think it is essential that the authors be more explicit with his goal, security analysis of sensors and threat Validation and countermeasures UNDER AN INTENTIONAL ELECTROMAGNETIC INTERFERENCE (IEMI) ATTACT.
ReThink as a new term the authors need a best definition and explanation of this term in the introduction and communicate to the readers the use of this term as a specific meaning in the rest of the article.
Line 730, “Here, we evaluate the effects of ReThink on the deviation” line 748 “6.1 Limits of Rethink”,” Consequently, it is imperative that the security analysis by ReThink should also be performed in these diverse domains”, “physical hardware vulnerabilities makes it difficult to 811 completely eliminate threats like ReThink by improving EMC standards”, “During the ReThink attack, the wanted EMI noise is injected into the input node”. Firstly, these phrases do not tell any information to the readers not expert in “ReThink”, furthermore, any interested person that do not read the file of ArXiv three times. Secon, these expressions are not proper of a research paper, because to mi knowledge this term is only use one time in a previous authors ArXiv file. The author there are not developed a term well understandable and defined in an intentional electromagnetic interference (IEMI) attack research expert community.
The familiarity with which the authors treat the term Rethink, which is their own, is poorly defined in the article, and is not an understandable term in the scientific world of this area, it is inappropriate for objective and subjective reasons. When the authors have a great and well knowledge background in this topic they will be able to talk about “ReThink” term with this familiarity. In my opinion it should be replaced in all sections and subsections titles, and used with caution, always with the citation of the ArXiv file, for example line 593, 604, 726, 727, 768.
With respect to the repetitive and specific use of the ReThink term. In conclusion, for the previous reasons, the content of this manuscript does not meet the minimum requirements to be a serious scientific article.
I have a little cosmetic issue with the EMI source icon (the devil), because it is naïve, but I show similar iconography in other articles of ACM. But I do not sow an evident solution, furthermore, I recommended to the editor that consider this question, because this is a part of the style of the authors and the Sensors editors.
There are no citation [1] in the introduction, This is connected with the word “ReThink” in the abstract and reference 1. I show reference [1] previous to abstract, I think this is not a reason for do not include in the introduction, and I think use as reference [1] is not the adequate order. Probably the editors know the correct form of solve this problem.
It is not correct introduce a personal (authors) developed definition “ReThink” in the abstract. In my knowledge it is forbidden use a citation in the abstract. Furthermore, rewrite in the abstract without the word “ReThink”, later in the introduction you can explain this word related with your work and cite properly, but not as the first reference because there other more important and generalist topics developed in the introduction.
Page 2 line 18 ” we propose three 17detection methods that are adaptable to diverse threat scenarios and the computational 18 resources available in inverters” the second part of the phrase is not understood.
Line 25 Reference 3 it is write in 2011 for a prediction of renewable energy in 2040 (29 years of range), I know this is a well know article, but find another more recent for that or similar extrapolation of energy.
Review the link at reference 4
Replace reference 6 wikimedia for a scholar or peer review text. Wikipedia it is not in this case proper for this topic.
This phrase “After performing a systematic security analysis of the PV inverters on real inverters and microgrid,” is needed to be corroborate with your published works and compared with works of other authors. .” we discover” only is referenced [16] of the year 2014, 11 years ago. This paragraph needed to be rewrite or include a substantial number of references because is a generalized phrase without the support of research publications.
Citations to references 14 and 15 are losses. Reference 14 it is incomplete.
Page 2, line 60 “First, the EMC is designed to cope with unintentional interference”, Could you explain what is the reason for consider intentional interferences, for example, a deliberate attack by EMI to the national RES net.
Page 2 line: “The EMC standard mainly” cite the related standards.
Include figure 1 under the ReThink explanation. Page 3 line 78.
What is ReThink in engineering terms?. It is a hardware?!, a program?!, an idea?!, a methodology of analysis?!. Be concrete with the explanation, because, from to now it is necessary read your Arxic article for understand the context. Now it is nededed the reference [1] and it is not included. I explain, “design ReThink (reveal the threat of EMI on inverters)” that is the consequence of applied Rethink? But there is not a comprehensive explanation of that.
Line 90, explain the type of generated EMI for , it is the same for all experiments (2 videos) or are at different frequencies, power or waveform signal.
I am showing the 2 https://tinyurl.com/ReThinkDemoVideos and they are very shelf-explanatory.
Due to the relevance of the videos, and if these could be accessible by other researches, with a Sensors link, it is needed numerate and explain the content of the videos, and identify equipment use for the EMI attack in each video case.
Now with a note page call, videos, I understand the previous topic “First, the EMC is designed to cope with unintentional interference,” because you develop a demonstrate experiments with deliberate EMI attack to RES inverters. For example ,with the EMI source antenna on a UVA (Unmanned aerial vehicle.), in this case an Unmanned combat aerial vehicle, because the goal is destroying the RES.
As I refer in the first paragraph of the review there is a problem with the explanation of the goal, be explicit, no use ReThink as a synonym of something similar to AN INTENTIONAL ELECTROMAGNETIC INTERFERENCE (IEMI) ATTACT
I think you will specify the goal of the article more clearly, the “MI threats” what the authors analyze are deliberate attacks not unintentional interference threat as an internal inverter EMI source or a external lightning bolt fell near the inverter.
Line 99 “Ti C2000 PV inverter” a reference is needed for the datasheet or manufacturer.
Line 100 “To the best of our knowledge, this is the first systematic work analyzing the impact 102 of EMI on PV inverters” specify that this EMI are external EMI to the inverters, produce by a deliberate attack.
Line 107 CPS , it needs to be defined (Cyber-Physical Systems (CPS)).
Line 141, references 30-39 are well done references for the topic IOT Devices, but it is an excessive number of references because are not related with inverters, and suppose the 10% of the total references. This is mi conclusion at this point. When a finish the manuscript lecturing I show two references of Xu, W into this topic outside or collateral to the PV inverters. Obviously, this is not ethical use of manuscript related scientific content. In this subsection I think that all the content is previously of later explained in the manuscript, Wop-amp-bases sensors”, “Clarke/Park transformation and deceive MPPT ” and problem with EMC standards obviously because do not contemplate the external attack possibility. recommended deleting this section and his references.
Line 258, Please identify the relation between “Rethink” ArXiv file and the 2.3 Threat model section, it is necessary write a phrase and the reference.
Line 277, probably there are mistakes in this phrase, because you refer to embedded voltage, and the section 3 is over embedded sensors, will be embedded voltage sensors.
Line 489, current serve, could be current sensor??
Line 593, The authors use the expression we evaluate, we first evaluate. In journal papers it is more common to use impersonal expressions, in some journal the personal expressions are practically forbidden.
The manuscript needs to be cut in those parts without technical content. Not only because read 36 page is boring, and loss the attention of the reader, if not because it avoids directly discussing what are his goals, what they have experienced and what results they have had, sometimes using the term Rethink as synonyms and other times holding writing circles over the central topics. For these reasons, some sections are repetitive and empty of technical or scientific content.
As an example of the previous reasoning the section Conclusions, 31 pages of previous text and 11 lines of conclusions. It is short, with bad redaction, and without the concrete and measured results of the manuscript.
The line “This paper presents a systematic analysis of the security vulnerabilities of PV inverters, 989 highlighting the impact of EMI on their voltage and current sensors” is the impact of intentional electromagnetic interference (IEMI) attack, not “impact of EMI” other “Threat posed by EMI” is a lie, is the posed by the impact of intentional electromagnetic interference (IEMI) attack. To the less in the conclusions, what is the main part of the article the content will be clear not ambiguous.
The phrase in the conclusions “Finally, we propose three effective detection methods based on different features”, but they are not mentioned.
References.
The author Xu, W has five references. I recommended the deletion of this three references do not related with the PV inverters. [36]. Wang, K.; Mitev, R.; Yan, C.; Ji, X.; Sadeghi, A.R.; Xu, W. {GhostTouch}: Targeted attacks on touchscreens without physical touch. 1090 In Proceedings of the 31st USENIX Security Symposium (USENIX Security 22), 2022, pp. 1543–1559. [39]. Jiang, Q.; Ji, X.; Yan, C.; Xie, Z.; Lou, H.; Xu,W. {GlitchHiker}: Uncovering Vulnerabilities of Image Signal Transmission with 1097
{IEMI}. In Proceedings of the 32nd USENIX Security Symposium (USENIX Security 23), 2023, pp. 7249–7266. [57]. Wang, K.; Mitev, R.; Yan, C.; Ji, X.; Sadeghi, A.R.; Xu, W. GhostTouch: Targeted Attacks on Touchscreens without Physical Touch. 1136 In Proceedings of the 31st USENIX Security Symposium (USENIX Security 22). USENIX Association, Boston, MA. https://www. 1137 usenix. org/conference/usenixsecurity22/presentation/wang-kai, 2022.
Round 2
Reviewer 3 Report
Comments and Suggestions for Authors
Good Work!
Reviewer 4 Report
Comments and Suggestions for Authors The revision of the article carried out by the authors has been good, and I consider that in this version the writing and graphics have improved significantly.